# Devito (v3.1.0): an embedded domain-specific language for finite differences and geophysical exploration

Mathias Louboutin[1], Michael Lange[2], Fabio Luporini[3], Navjot Kukreja[3], Philipp A. Witte[1], Felix J. Herrmann[1], Paulius Velesko[3], and Gerard J. Gorman[3]

[1]School of Computational Science and Engineering, Georgia Institute of Technology, Atlanta, USA
[2]ECMWF, UK
[3]Earth Science and Engineering, Imperial College London, London, UK

*Correspondence to:* Mathias Louboutin (mlouboutin3@gatech.edu)

**Abstract.**

We introduce Devito, a new domain-specific language for implementing high-performance finite difference partial differential equation solvers. The motivating application is exploration seismology where methods such as Full-Waveform Inversion and Reverse-Time Migration are used to invert terabytes of seismic data to create images of the earth's subsurface. Even using modern supercomputers, it can take weeks to process a single seismic survey and create a useful subsurface image. The computational cost is dominated by the numerical solution of wave equations and their corresponding adjoints. Therefore, a great deal of effort is invested in aggressively optimizing the performance of these wave-equation propagators for different computer architectures. Additionally, the actual set of partial differential equations being solved and their numerical discretization is under constant innovation as increasingly realistic representations of the physics are developed, further ratcheting up the cost of practical solvers. By embedding a domain-specific language within Python and making heavy use of SymPy, a symbolic mathematics library, we make it possible to develop finite difference simulators quickly using a syntax that strongly resembles the mathematics. The Devito compiler reads this code and applies a wide range of analysis to generate highly optimized and parallel code. This approach can reduce the development time of a verified and optimized solver from months to days.

## 1 Introduction

Large-scale inversion problems in exploration seismology constitute some of the most computationally demanding problems in industrial and academic research. Developing computationally efficient solutions for applications such as seismic inversion requires expertise ranging from theoretical and numerical methods for partial differential equation (PDE) constrained optimization to low-level performance optimization of PDE solvers. Progress in this area is often limited by the complexity and cost of developing bespoke wave propagators (and their discrete adjoints) for each new inversion algorithm or formulation of wave physics. Traditional software engineering approaches often lead developers to make critical choices regarding the numerical discretization before manual performance optimization for a specific target architecture and making it ready for production. This workflow of bringing new algorithms into production, or even to a stage that they can be evaluated on realistic datasets can take many person-months or even person-years. Furthermore, it leads to mathematical software that is not easily ported,

maintained or extended. In contrast, the use of high-level abstractions and symbolic reasoning provided by domain-specific languages (DSL) can significantly reduce the time it takes to implement and verify individual operators for use in inversion frameworks, as has already been shown for the finite element method (Logg et al., 2012; Rathgeber et al., 2015; Farrell et al., 2013).

State-of-the-art seismic imaging is primarily based upon explicit finite difference schemes due to their relative simplicity and ease of implementation (Virieux, 1986; Symes, 2015a; Weiss and Shragge, 2013). When considering how to design a DSL for explicit finite difference schemes, it is useful to recognize the algorithm as being primarily a sub-class of stencil algorithms or polyhedral computation (Henretty et al., 2013; Andreolli et al., 2015; Yount, 2015). However, stencil compilers lack two significant features required to develop a DSL for finite differences: symbolic computational support required to express finite difference discretizations at a high level, enabling these expressions to be composed and manipulated algorithmically; support for algorithms that are not stencil-like, such as source and receiver terms that are both sparse and unaligned with the finite difference grid. Therefore, the design aims behind the Devito DSL can be summarized as:

- create a high-level mathematical abstraction for programming finite differences to enable composability and algorithmic optimization,

- insofar as possible use existing compiler technologies to optimize the affine loop nests of the computation, which account for most of the computational cost,

- develop specific extensions for other parts of the computation that are non-affine (e.g., source and receiver terms).

The first of these aims is primarily accomplished by embedding the DSL in *Python* and leveraging the symbolic mathematics package Sympy (Meurer et al., 2017). From this starting point, an abstract syntax tree is generated and standard compiler algorithms can be employed to either generate optimized and parallel C code or to write code for a stencil DSL - which itself will be passed to the next compiler in the chain. The fact that this can be all performed just-in-time (JIT) means that a combination of static and dynamic analysis can be used to generate optimized code. However, in some circumstances, one might also choose to compile offline.

The use of symbolic manipulation, code generation, and just-in-time compilation allows the definition of individual wave propagators for inversion in only a few lines of *Python* code, while aspects such as varying the problem discretization become as simple as changing a single parameter in the problem specification, for example, changing the order of the spatial discretization (Louboutin et al., 2017a). This article explains *what* can be accomplished with Devito, showing how to express real-life wave propagators as well as their integration within larger workflows typical of seismic exploration, such as the popular Full-Waveform Inversion (FWI) and Reverse-Time Migration (RTM) methods. The Devito compiler, and in particular *how* the user-provided SymPy equations are translated into high-performance C, are also briefly summarized, although for a complete description the interested reader should refer to Luporini et al. (2018).

The remainder of this paper is structured as follows: first, we provide a brief history of optimizing compilers, DSL and existing wave equation seismic frameworks. Next, we highlight the core features of Devito and describe the implementation of

the featured wave equation operators in Sec. 3. We outline the seismic inversion theory in Sec. 4. Code verification and analysis of accuracy in Sec. 5 is followed by a discussion of the propagators computational performance in Sec. 6. We conclude by presenting a set of realistic examples such as seismic inversion and computational fluid dynamics and a discussion of future work.

## 2   Background

Improving the runtime performance of a critical piece of code on a particular computing platform is a non-trivial task that has received significant attention throughout the history of computing. The desire to automate the performance optimization process itself across a range of target architectures is not new either, although it is often met with skepticism. Even the very first compiler, A0 (Hopper, 1952), was received with resistance, as best summarized in the following quote: *"Dr. Hopper believes,…, that the result of a compiling technique should be a routine just as efficient as a hand tailored routine. Some others do not completely agree with this. They feel the machine-made routine can approach hand tailored coding, but they believe there are "tricks of the trade" that apply to various special cases that a computer cannot be expected to utilize."* (Jones, 1954). Given the challenges of porting optimized codes to a wide range of rapidly evolving computer architectures, it seems natural to raise again the layer of abstraction and use compiler techniques to replace much of the manual labor.

Community acceptance of these new "automatic coding systems" began when concerns about the performance of the generated code were addressed by the first "optimizing compiler", FORTRAN, released in 1957 – which not only translated code from one language to another but also ensured that the final code performed at least as good as a hand-written low-level code (Backus, 1978). Since then, as program and hardware complexity rose, the same problem has been solved over and over again, each time by the introduction of higher levels of abstractions. The first high-level languages and compilers were targeted at solving a large variety of problems and hence were restricted in the kind of optimizations they could leverage. As these generic languages became common-place and the need for further improvement in performance was felt, restricted languages focusing on smaller problem domains were developed that could leverage more "tricks of the trade" to optimize performance. This led to the proliferation of DSLs for broad mathematical domains or sub-domains, such as APL (Iverson, 1962), Mathematica, Matlab®or R.

In addition to these relatively general mathematical languages, more specialized frameworks targeting the automated solution of PDEs have long been of interest (Cárdenas and Karplus, 1970; Umetani, 1985; Cook Jr, 1988; Van Engelen et al., 1996). More recent examples not only aim to encapsulate the high-level notation of PDEs, but are often centered around a particular numerical method. Two prominent contemporary projects based on the finite element method (FEM), FEniCS (Logg et al., 2012) and Firedrake (Rathgeber et al., 2015), both implement a common DSL, UFL (Alnæs et al., 2014), that allows the expression of variational problems in weak form. Multiple DSLs to express stencil-like algorithms have also emerged over time, including (Henretty et al., 2013; Zhang and Mueller, 2012; Christen et al., 2011; Unat et al., 2011; Köster et al., 2014; Membarth et al., 2012; Osuna et al., 2014; Tang et al., 2011; Bondhugula et al., 2008; Yount, 2015). Other stencil DSLs have been developed with the objective of solving PDEs using finite differences (Hawick and Playne, 2013; Arbona et al., 2017;

Jacobs et al., 2016). However, in all cases their use in seismic imaging problems (or even more broadly in science and engineering) has been limited by a number of factors other than technology inertia. Firstly, they only raise the abstraction to the level of polyhedral-like (affine) loops. As they do not generally use a symbolic mathematics engine to write the mathematical expressions at a high-level, developers must still write potentially complex numerical kernels in the target low-level programming language. For complex formulations, this process can be time-consuming and error prone, as hand-tuned codes for wave propagators can reach thousands of lines of code. Secondly, most DSLs rarely offer enough flexibility for extension beyond their original scope (e.g. sparse operators for source terms and interpolation) making it difficult to work the DSL into a more complex science/engineering workflow. Finally, since finite difference wave propagators only form part of the over-arching PDE constrained (wave equation) optimization problem, composability with external packages, such as the *SciPy* optimization toolbox, is a key requirement that is often ignored by self-contained standalone DSLs. The use of a fully embedded *Python* DSL, on the other hand, allows users to leverage a variety of higher-level optimization techniques through a rich variety of software packages provided by the scientific *Python* ecosystem.

Moreover, several computational frameworks for seismic imaging exist, although they provide varying degrees of abstraction and are typically limited to a single representation of the wave equation. IWAVE (Symes et al., 2011; Symes, 2015b; Sun and Symes, 2010; Symes, 2015a), although not a DSL, provides a high-level of abstraction and a mathematical framework to abstract the algebra related to the wave-equation and its solution. IWAVE provides a rigorous mathematical abstraction for linear operations and vector representations including Hilbert space abstraction for norms and distances. However, its C++ implementation limits the extensibility of the framework to new wave-equations. Other software frameworks, such as Madagascar (Fomel et. al, 2013), offer a broad range of applications. Madagascar is based on a set of subroutines for each individual problem and offers modelling and imaging operators for multiple wave-equations. However, the lack of high-level abstraction restricts its flexibility and interfacing with high-level external software (i.e *Python* , *Java*). The subroutines are also mostly written in C/Fortran and limit the architecture portability.

## 3 Symbolic definition of finite difference stencils with Devito

In general, the majority of the computational workload in wave-equation based seismic inversion algorithms arises from computing solutions to discrete wave equations and their adjoints. There are a wide range of mathematical models used in seismic imaging that approximate the physics to varying degrees of fidelity. To improve development and innovation time, including code verification, we describe the use of the symbolic finite difference framework Devito to create a set of explicit matrix-free operators of arbitrary spatial discretization order. These operators are derived for example from the acoustic wave equation

$$m(x)\frac{\partial^2 u(t,x)}{\partial t^2} - \Delta u(t,x) = q(t,x), \tag{1}$$

where $m(x) = \frac{1}{c(x)^2}$ is the squared slowness with $c(x)$ the spatially dependent speed of sound, symbol $\Delta u(t,x)$ denotes the Laplacian of the wavefield $u(t,x)$ and $q(t,x)$ is a source usually located at a single location $x_s$ in space ($q(t,x) = f(t)\delta(x_s)$). This formulation will be used as running example throughout the section.

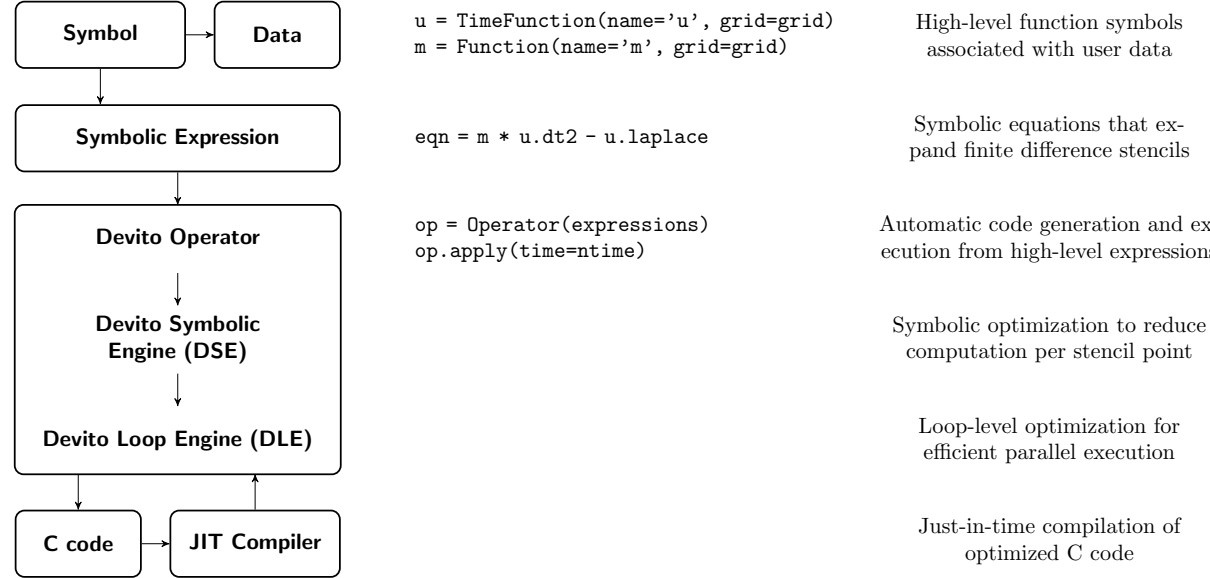

**Figure 1.** Overview of the Devito architecture and the associated example workflow. Devito's top-level API allows users to generate symbolic stencil expressions using data-carrying function objects that can be used for symbolic expressions via *SymPy* . From this high-level definition, an operator then generates, compiles and executes high-performance C code.

### 3.1 Code generation - an overview

Devito aims to combine performance benefits of dedicated stencil frameworks (Bondhugula et al., 2008; Tang et al., 2011; Henretty et al., 2013; Yount, 2015) with the expressiveness of symbolic PDE-solving DSLs (Logg et al., 2012; Rathgeber et al., 2015) through automated code generation and optimization from high-level symbolic expressions of the mathematics.

Thus, the primary design objectives of the Devito DSL are to allow users to define explicit finite difference operators for (time-dependent) PDEs in a concise symbolic manner and provide an API that is flexible enough to fully support realistic scientific use cases. To this end, Devito offers a set of symbolic classes that are fully compatible with the general-purpose symbolic algebra package *SymPy* that enables users to derive discretized stencil expressions in symbolic form. As we show in Fig. 1, the primary symbols in such expressions are associated with user data that carry domain-specific meta-data information to be used

by the compiler engine (e.g. dimensions, data type, grid). The discretized expressions form an abstract operator definition that Devito uses to generate low-level C code (C99) and OpenMP at runtime. The encapsulating `Operator` object can be used to execute the generated code from within the *Python* interpreter making Devito natively compatible with the wide range of tools available in the scientific *Python* software stack. We manage memory using our own allocators (e.g. to enforce alignment and NUMA optimizations) and therefore we also take control over freeing memory. We wrap everything with the NumPy array

API to ensure interoperability with other modules that use NumPy.

A Devito `Operator` takes as input a collection of symbolic expressions and progressively lowers the symbolic representation to semantically equivalent C code. The code generation process consists of a sequence of compiler passes during which

multiple automated performance-optimization techniques are employed. These can be broadly categorised into two classes and are performed by distinct sub-packages:

- **Devito Symbolic Engine (DSE):** Symbolic optimization techniques, such as Common Sub-expression Elimination (CSE), factorization and loop-invariant code motion are utilized to reduce the number of floating point operations (flops) performed within the computational kernel (Luporini et al., 2015). These optimization techniques are inspired by *SymPy* but are custom implemented in Devito and do not rely on *SymPy* implementation of CSE for example.

- **Devito Loop Engine (DLE):** Well-known loop optimization techniques, such as explicit vectorization, thread-level parallelization and loop blocking with auto-tuned block sizes are employed to increase the cache utilization and thus memory bandwidth utilization of the kernels.

A complete description of the compilation pipeline is provided in Luporini et al. (2018).

## 3.2 Discrete function symbols

The primary user-facing API of Devito allows the definition of complex stencil operators from a concise mathematical notation. For this purpose, Devito relies strongly on *SymPy* (Devito 3.1.0 depends upon SymPy 1.1 and all dependency versions are specified in Devito's requirements file). Devito provides two symbolic object types that mimic *SymPy* symbols, enabling the construction of stencil expressions in symbolic form:

- **Function:** The primary class of symbols provided by Devito behaves like `sympy.Function` objects, allowing symbolic differentiation via finite difference discretization and general symbolic manipulation through *SymPy* utilities. Symbolic function objects encapsulate state variables (parameters and solution of the PDE) in the operator definition and associated user data (function value) with the represented symbol. The meta-data, such as grid information and numerical type, which provide domain-specific information to the Devito compiler are also carried by the `sympy.Function` object.

- **Dimension:** Each `sympy.Function` object defines an iteration space for stencil operations through a set of `Dimension` objects that are used to define and generate the corresponding loop structure from the symbolic expressions.

In addition to `sympy.Function` and `Dimension` symbols, Devito supplies the construct `Grid`, which encapsulates the definition of the computational domain and defines the discrete shape (number of grid points, grid spacing) of the function data. The number of spatial dimensions is hereby derived from the shape of the `Grid` object and inherited by all `Function` objects, allowing the same symbolic operator definitions to be used for two and three-dimensional problem definitions. As an example, a two-dimensional discrete representation of the square slowness of an acoustic wave $m[x, y]$ inside a 5 by 6 grid points domain can be created as a symbolic function object as illustrated in Fig. 2.

It is important to note here that $m[x, y]$ is constant in time, while the discrete wavefield $u[t, x, y]$ is time-dependent. Since time is often used as the stepping dimension for buffered stencil operators, Devito provides an additional function type

```
>>> grid = Grid(shape=(5, 6))
>>> m = Function(name='m',
    grid=grid)

>>> m
m(x, y)

>>> m.data.shape
(5, 6)
```

**Figure 2.** Defining a Devito `Function` on a `Grid`.

`TimeFunction`, which automatically adds a special `TimeDimension` object to the list of dimensions. `TimeFunction` objects derive from `Function` with an extra time dimension and inherit all the symbolic properties. The creation of a `TimeFunction` requires the same parameters as a `Function`, with an extra optional `time_order` property that defines the discretization order for the time dimension and an integer `save` parameter that defines the size of the time axis when
the full time history of the field is stored in memory. In the case of a buffered time dimension `save` is equal to `None` and the size of the buffered dimension is automatically inferred from the `time_order` value. As an example, we can create an equivalent symbolic representation of the wavefield as `u = TimeFunction(name='u', grid=grid)`, which is denoted symbolically as `u(t, x, y)`.

### 3.2.1 Spatial discretization

The symbolic nature of the function objects allows the automatic derivation of discretized finite difference expressions for derivatives. Devito `Function` objects provide a set of shorthand notations that allow users to express, for example, $\frac{d\boldsymbol{u}[t,x,y,z]}{dx}$ as `u.dx` and $\frac{d^2\boldsymbol{u}[t,x,y,z]}{dx^2}$ as `u.dx2`. Moreover, the discrete Laplacian, defined in three dimensions as $\Delta\boldsymbol{u}[t,x,y,z] = \frac{d^2\boldsymbol{u}[t,x,y,z]}{dx^2} + \frac{d^2\boldsymbol{u}[t,x,y,z]}{dy^2} + \frac{d^2\boldsymbol{u}[t,x,y,z]}{dz^2}$ can be expressed in shorthand simply as `u.laplace`. The shorthand expression `u.laplace` is agnostic to the number of spatial dimensions and may be used for two or three-dimensional problems.

The discretization of the spatial derivatives can be defined for any order. In the most general case, we can write the spatial discretization in the $x$ direction of order $k$ (and equivalently in the $y$ and $z$ direction) as:

$$\frac{\partial^2\boldsymbol{u}[t,x,y,z]}{\partial x^2} = \frac{1}{h_x^2}\sum_{j=0}^{\frac{k}{2}}\left[\alpha_j\left(\boldsymbol{u}[t,x+jh_x,y,z]+\boldsymbol{u}[t,x-jh_x,y,z]\right)\right], \tag{2}$$

where $h_x$ is the discrete grid spacing for the dimension $x$, the constants $\alpha_j$ are the coefficients of the finite difference scheme and the spatial discretization error is of order $O(h_x^k)$.

### 3.2.2 Temporal discretization

We consider here a second-order time discretization for the acoustic wave equation, as higher order time discretization requires us to rewrite the PDE (Seongjai Kim, 2007). The discrete second-order time derivative with this scheme can be derived from the Taylor expansion of the discrete wavefield $\boldsymbol{u}(t,x,y,z)$ as:

$$\frac{d^2\boldsymbol{u}[t,x,y,z]}{dt^2} = \frac{\boldsymbol{u}[t+\Delta t,x,y,z] - 2\boldsymbol{u}[t,x,y,z] + \boldsymbol{u}[t-\Delta t,x,y,z]}{\Delta t^2}. \tag{3}$$

In this expression, $\Delta t$ is the size of a discrete time step. The discretization error is $O(\Delta t^2)$ (second order in time) and will be verified in Sec. 5.

Following the convention used for spatial derivatives, the above expression can be automatically generated using the short-hand expression `u.dt2`. Combining the temporal and spatial derivative notations, and ignoring the source term $q$, we can
now define the wave propagation component of Eq. 1 as a symbolic expression via `Eq(m * u.dt2 - u.laplace, 0)` where `Eq` is the *SymPy* representation of an equation. In the resulting expression, all spatial and temporal derivatives are expanded using the corresponding finite difference terms. To define the propagation of the wave in time, we can now rearrange the expression to derive a stencil expression for the forward stencil point in time, $\boldsymbol{u}(t+\Delta t,x,y,z)$, denoted by the shorthand expression `u.forward`. The forward stencil corresponds to the explicit Euler time-stepping that updates the next time-step
`u.forward` from the two previous ones `u` and `u.backward` (Eq. 4). We use the *SymPy* utility `solve` to automatically derive the explicit time-stepping scheme, as shown in Fig. 3 for the second order in space discretization.

$$\boldsymbol{u}[t+\Delta t,x,y,z] = 2\boldsymbol{u}[t,x,y,z] - \boldsymbol{u}[t-\Delta t,x,y,z] + \frac{\Delta t^2}{\boldsymbol{m}[x,y,z]} \Delta\boldsymbol{u}[t,x,y,z]. \tag{4}$$

The iteration over time to obtain the full solution is then generated by the Devito compiler from the time dimension information. Solving the wave-equation with the above explicit Euler scheme is equivalent to a linear system $\mathbf{A}(\boldsymbol{m})\boldsymbol{u} = \boldsymbol{q}_s$ where
the vector $\boldsymbol{u}$ is the discrete wavefield solution of the discrete wave-equation, $\boldsymbol{q}_s$ is the source term and $\mathbf{A}(\boldsymbol{m})$ is the matrix representation of the discrete wave-equation. From Eq. 4 we can see that the matrix $\mathbf{A}(\boldsymbol{m})$ is a lower triangular matrix that reflects the time-marching structure of the stencil. Simulation of the wavefield is equivalent to a forward substitution (solve row by row from the top) on the lower triangular matrix $\mathbf{A}(\boldsymbol{m})$. Since we do not consider complex valued PDEs, the adjoint of $\mathbf{A}(\boldsymbol{m})$ is equivalent to its transpose denoted as $\mathbf{A}^\top(\boldsymbol{m})$ and is an upper triangular matrix. The solution $\boldsymbol{v}$ of the discrete adjoint
wave-equation $\mathbf{A}(\boldsymbol{m})^\top\boldsymbol{v} = \boldsymbol{q}_a$ for an adjoint source $\boldsymbol{q}_a$ is equivalent to a backward substitution (solve from the bottom row to top row) on the upper triangular matrix $\mathbf{A}(\boldsymbol{m})^\top$ and is simulated backward in time starting from the last time-step. These matrices are never explicitly formed, but are instead matrix free operators with implicit implementation of the matrix-vector product, $\boldsymbol{u} = \mathbf{A}(\boldsymbol{m})^{-1}\boldsymbol{q}_s$ as a forward stencil. The stencil for the adjoint wave-equation in this self-adjoint case would simply be obtained with `solve(eqn, u.backward)` and the compiler will detect the backward-in-time update.

```
>>> from sympy import Eq, solve, init_printing, pprint
>>> init_printing(use_latex=True)
>>> from devito import Function, TimeFunction, Grid

>>> grid = Grid(shape=(5, 5))
>>> u = TimeFunction(name='u', grid=grid, space_order=2, time_order=2)
>>> m = Function(name='m',grid=grid)

>>> eqn = Eq(m * u.dt2 - u.laplace)
>>> stencil = solve(eqn, u.forward)[0]
>>> pprint(Eq(u.forward, stencil))
```

Produces output equivalent to:

$$
\begin{aligned}
u(t + s, x, y) = {} & 2u(t, x, y) - u(t - s, x, y) \\
& \left. \begin{array}{l}
- \dfrac{2s^2 u(t, x, y)}{h_y^2 m(x, y)} + \dfrac{s^2 u(t, x, y - h_y)}{h_y^2 m(x, y)} + \dfrac{s^2 u(t, x, y + h_y)}{h_y^2 m(x, y)} \\[2ex]
- \dfrac{2s^2 u(t, x, y)}{h_x^2 m(x, y)} + \dfrac{s^2 u(t, x - h_x, y)}{h_x^2 m(x, y)} + \dfrac{s^2 u(t, x + h_x, y)}{h_x^2 m(x, y)}
\end{array} \right\} \dfrac{\Delta t^2 \Delta u}{m(x, y)}
\end{aligned}
$$

**Figure 3.** Example code defining the two-dimensional wave equation without damping using Devito symbols and symbolic processing utilities from *SymPy* . Assuming $h_x = \Delta x$, $h_y = \Delta y$ and $s = \Delta t$ the output is equivalent to Eq. 1 without the source term $\boldsymbol{q}_s$.

### 3.2.3 Boundary conditions

The field recorded data is measured on a wavefield that propagates in an infinite domain. However, solving the wave equation in a discrete infinite domain is not feasible with finite differences. In order to mimic an infinite domain, Absorbing Boundary Conditions (ABC) or Perfectly Matched Layers (PML) are necessary (Clayton and Engquist, 1977). These two methods allow the approximation of the wavefield as it is in an infinite medium by damping and absorbing the waves within an extra layer at the limit of the domain to avoid unnatural reflections from the edge of the discrete domain.

The least computationally expensive method is the Absorbing Boundary Condition that adds a single damping mask in a finite layer around the physical domain. This absorbing condition can be included in the wave-equation as:

$$
\boldsymbol{m}[x,y,z]\frac{d^2\boldsymbol{u}[t,x,y,z]}{dt^2} - \Delta\boldsymbol{u}[t,x,y,z] + \boldsymbol{\eta}[x,y,z]\frac{d\boldsymbol{u}[t,x,y,z]}{dt} = 0. \tag{5}
$$

The $\boldsymbol{\eta}[x,y,z]$ parameter is equal to 0 inside the physical domain and increasing from inside to outside within the damping layer. The dampening parameter $\boldsymbol{\eta}$ can follow a linear or exponential curve depending on the frequency band and width of the dampening layer. For methods based on more accurate modelling, for example in simulation-based acquisition design (Liu and Fomel, 2011; Wason et al., 2017; Naghizadeh and Sacchi, 2009; Kumar et al., 2015), a full implementation of the PML will be necessary to avoid weak reflections from the domain limits.

### 3.2.4 Sparse point interpolation

Seismic inversion relies on data fitting algorithms, hence we need to support sparse operations such as source injection and wavefield ($\boldsymbol{u}[t,x,y,z]$) measurement at arbitrary grid locations. Both operations occur at sparse domain points, which do not necessarily align with the logical cartesian grid used to compute the discrete solution $\boldsymbol{u}(t,x,y,z)$. Since such operations are not captured by the finite differences abstractions for implementing PDEs, Devito implements a secondary high-level representation of sparse objects (Lange et al., 2017) to create a set of *SymPy* expressions that perform polynomial interpolation within the containing grid cell from pre-defined coefficient matrices.

The necessary expressions to perform interpolation and injection are automatically generated through a dedicated symbol type, `SparseFunction`, which associates a set of coordinates with the symbol representing a set of non-aligned points. For examples, the syntax `p.interpolate(expr)` provided by a `SparseFunction p` will generate a symbolic expressions that interpolates a generic expression `expr` onto the sparse point locations defined by p, while `p.inject(field, expr)` will evaluate and add `expr` to each enclosing point in `field`. The generated *SymPy* expressions are passed to Devito `Operator` objects alongside the main stencil expression to be incorporated into the generated C kernel code. A complete setup of the acoustic wave equation with absorbing boundaries, injection of a source function and measurement of wavefields via interpolation at receiver locations can be found in Sec. 4.2.

## 4 Seismic modeling and inversion

Seismic inversion methods aim to reconstruct physical parameters or an image of the earth's subsurface from multi-experiment field measurements. For this purpose, a wave is generated at the ocean surface that propagates through to the subsurface and creates reflections at the discontinuities of the medium. The reflected and transmitted waves are then captured by a set of hydrophones that can be classified as either moving receivers (cables dragged behind a source vessel) or static receivers (ocean bottom nodes or cables). From the acquired data, physical properties of the subsurface such as wave speed or density can be reconstructed by minimizing the misfit between the recorded measurements and the numerically modelled seismic data.

### 4.1 Full-Waveform Inversion

Recovering the wave speed of the subsurface from surface seismic measurements is commonly cast into a non-linear optimization problem called full-waveform inversion (FWI). The method aims at recovering an accurate model of the discrete wave velocity, $\boldsymbol{c}$, or alternatively, the square slowness of the wave, $\boldsymbol{m} = \frac{1}{c^2}$(not an overload), from a given set of measurements of the pressure wavefield $\boldsymbol{u}$. Lions (1971); Tarantola (1984); Virieux and Operto (2009); Haber et al. (2012) shows that this can be expressed as a PDE-constrained optimization problem. After elimination of the PDE constraint, the reduced objective function is defined as:

$$\underset{\boldsymbol{m}}{\text{minimize}} \quad \Phi_s(\boldsymbol{m}) = \frac{1}{2}\|\mathbf{P}_r\boldsymbol{u} - \boldsymbol{d}\|_2^2 \quad \text{with:} \quad \boldsymbol{u} = \mathbf{A}(\boldsymbol{m})^{-1}\mathbf{P}_s^T\boldsymbol{q}_s, \tag{6}$$

where $\mathbf{P}_r$ is the sampling operator at the receiver locations, $\mathbf{P}_s^T$ ($^T$ is the transpose or adjoint) is the injection operator at the source locations, $\mathbf{A}(\boldsymbol{m})$ is the operator representing the discretized wave equation matrix, $\boldsymbol{u}$ is the discrete synthetic pressure wavefield, $\boldsymbol{q}_s$ is the corresponding pressure source and $\boldsymbol{d}$ is the measured data. While we consider the acoustic isotropic wave equation for simplicity here, in practice, multiple implementations of the wave equation operator $\boldsymbol{A}(\boldsymbol{m})$ are possible depending

on the choice of physics. In the most advanced case, $\boldsymbol{m}$ would not only contain the square slowness but also anisotropic or orthorhombic parameters.

To solve this optimization problem with a gradient-based method, we use the adjoint-state method to evaluate the gradient (Plessix, 2006; Haber et al., 2012):

$$\nabla\Phi_s(\boldsymbol{m}) = \sum_{\boldsymbol{t}=1}^{n_t} \boldsymbol{u}[t]\boldsymbol{v}_{tt}[t] = \mathbf{J}^T \delta\boldsymbol{d}_s, \tag{7}$$

where $n_t$ is the number of computational time steps, $\delta\boldsymbol{d}_s = (\mathbf{P}_r\boldsymbol{u} - \boldsymbol{d})$ is the data residual (difference between the measured data and the modeled data), $\mathbf{J}$ is the Jacobian operator and $\boldsymbol{v}_{tt}$ is the second-order time derivative of the adjoint wavefield that solves:

$$\mathbf{A}^T(\boldsymbol{m})\boldsymbol{v} = \mathbf{P}_r^T \delta\boldsymbol{d}_s. \tag{8}$$

The discretized adjoint system in Eq. 8 represents an upper triangular matrix that is solvable by modelling wave propagation

backwards in time (starting from the last time step). The adjoint state method, therefore, requires a wave equation solve for both the forward and adjoint wavefields to compute the gradient. An accurate and consistent adjoint model for the solution of the optimization problem is therefore of fundamental importance.

### 4.2 Acoustic forward modelling operator

We consider the acoustic isotropic wave-equation parameterized in terms of slowness $\boldsymbol{m}[x, y, z]$ with zero initial conditions

assuming the wavefield does not have any energy before zero time. We define an additional dampening term to mimic an infinite domain (see Sec. 3.2.3). At the limit of the domain, the zero Dirichlet boundary condition is satisfied as the solution is considered to be fully damped at the limit of the computational domain. The PDE is defined in Eq. 5. Figure 4 demonstrates the complete set up of the acoustic wave equation with absorbing boundaries, injection of a source function and sampling wavefields at receiver locations. The shape of the computational domain is hereby provided by a utility object `model`, while

the damping term $\eta\frac{d\boldsymbol{u}[x,y,z,t]}{dt}$ is implemented via a utility symbol `eta` defined as a `Function` object. It is important to note that the discretization order of the spatial derivatives is passed as an external parameter `order` and carried as meta-data by the wavefield symbol `u` during construction, allowing the user to freely change the underlying stencil order.

The main (PDE) stencil expression to update the state of the wavefield is derived from the high-level wave equation expression `eqn = u.dt2 - u.laplace + damp*u.dt` using *SymPy* utilities as demonstrated before in Fig. 3. Additional

30 expressions for the injection of the wave source via the `SparseFunction` object `src` are then generated for the forward

```
def forward(model, source, receiver,
    space_order=2):
  m, eta = model.m, model.damp
  # Allocate wavefield and auxiliary fields
  u = TimeFunction(name='u', grid=model.grid,
      time_order=2,
                  space_order=space_order)

  # Derive stencil from symbolic equation
  eqn = m * u.dt2 - u.laplace + eta * u.dt
  stencil = solve(eqn, u.forward)
  update_u = Eq(u.forward, stencil)

  # Source injection and receiver interpolation
  src = source.inject(field=u.forward,
      expr=src * dt**2 / m)
  rec = receiver.interpolate(expr=u)

  op = Operator([update_u] + src + rec,
      subs=model.spacing_map)
```

**Figure 4.** Example definition of a forward operator.

wavefield, where the source time signature is discretized onto the computational grid via the symbolic expression `src *`
`dt**2 / m`. The weight $\frac{dt^2}{m}$ is derived from rearranging the discretized wave equation with a source as a right-hand-side
similarly to the Laplacian in Eq. 4. A similar expression to interpolate the current state of the wavefield at the receiver locations
(measurement points) is generated through the `receiver` symbol. The combined list of stencils, a sum in Python that adds

the different expressions that update the wavefield at the next time step, inject the source and interpolate at the receivers, is
then passed to the `Operator` constructor alongside a definition of the spatial and temporal spacing $h_x, h_y, h_z, \Delta t$ provided
by the `model` utility. Devito then transforms this list of stencil expressions into loops (inferred from the symbolic Functions),
replaces all necessary constants by their values if requested, prints the generated C code and compiles it. The operator is finally
callable in Python with `op.apply()`.

A more detailed explanation of the seismic setup and parameters such as the source and receiver terms in Fig. 4 is covered
in Louboutin et al. (2017b).

### 4.3 Discrete adjoint wave-equation and FWI gradient

To create the adjoint that pairs with the above forward modeling propagator we can make use of the fact that the isotropic
acoustic wave equation is self-adjoint. This entails that for the implementation of the forward wave equation `eqn`, shown

in Fig. 5, only the sign of the damping term needs to be inverted, as the dampening time-derivative has to be defined in the
direction of propagation ($\frac{\partial}{\partial n(t)}$). For the PDE stencil, however, we now rearrange the stencil expression to update the backward
wavefield from the two next time steps as $\boldsymbol{v}[t-\Delta t, x, y, z] = f(\boldsymbol{v}[t, x, y, z], \boldsymbol{v}[t+\Delta t, x, y, z])$. Moreover, the role of the sparse

```
def adjoint(model, adj_src, adj_rec,
    space_order=2):
  m, eta = model.m, model.damp
  # Allocate wavefield and auxiliary fields
  v = TimeFunction(name='v', grid=model.grid,
              time_order=2,
                  space_order=space_order)

  # Derive stencil from symbolic equation
  eqn = m * v.dt2 - v.laplace - eta * v.dt
  stencil = solve(eqn, v.backward)
  update_v = Eq(u.backward, stencil)

  # Receiver injection and adj-source
      interpolation
  src_a = adj_src.inject(field=v.backward,
                  expr=rec * dt**2 / m)
  rec_a = adj_rec.interpolate(expr=v)

  op = Operator([update_v] + src_a + rec_a,
            subs=model.spacing_map)
```

**Figure 5.** Example definition of an adjoint operator.

```
def gradient(model, u, adj_src, space_order=2):
  m, eta = model.m, model.damp
  # Allocate wavefield and auxiliary fields
  v = TimeFunction(name='v', grid=model.grid,
          time_order=2,
              space_order=space_order)
  grad = Function(name='grad', grid=model.grid)

  # Derive stencil from symbolic equation
  eqn = m * v.dt2 - v.laplace - eta * v.dt
  stencil = solve(eqn, v.backward)
  update_v = Eq(u.backward, stencil)

  # Receiver injection and gradient update
  src_a = adj_src.inject(field=v.backward,
                  expr=rec * dt**2 / m)
  update_grad = Eq(grad, grad - u * v.dt2)

  op = Operator([update_v] + src_a +
      update_grad,
            subs=model.spacing)
```

**Figure 6.** Example definition of a gradient operator.

point symbols has changed (Eq. 8), so that we now inject time-dependent data at the receiver locations (`adj_src`), while sampling the wavefield at the original source location (`adj_rec`).

Based on the definition of the adjoint operator, we can now define a similar operator to update the gradient according to Eq. 7. As shown in Fig. 6, we can replace the expression to sample the wavefield at the original source location with an accumulative
update of the gradient field `grad` via the symbolic expression `Eq(grad, grad - u * v.dt2)`.

To compute the gradient, the forward wavefield at each time step must be available which leads to significant memory requirements. Many methods exist to tackle this memory issue, but all come with their advantages and disadvantages. For instance, we implemented optimal checkpointing with the library Revolve (Griewank and Walther, 2000) in Devito to drastically reduce the memory cost by only saving a partial time history and recomputing the forward wavefield when needed (Kukreja
et al., 2018). The memory reduction comes at an extra computational cost as optimal checkpointing requires $log(n_t) + 2$ extra PDE solves. Another method is boundary wavefield reconstruction (McMechan, 1983; Mittet, 1994; Raknes and Weibull, 2016) that saves the wavefield only at the boundary of the model, but still requires us to recompute the forward wavefield during the back-propagation. This boundary method has a reduced memory cost but necessitates the computation of the forward wavefield twice (one extra PDE solve), once to get the data than a second time from the boundary values to compute the gradient.

**4.4   FWI using Devito operators**

At this point, we have a forward propagator to model synthetic data in Fig. 4, the adjoint propagator for Eq. 8 and the FWI gradient of Eq. 7 in Fig. 6. With these three operators, we show the implementation of the FWI objective and gradient with

```
def fwi_gradient(model, op_fwd, op_grad):
    """
    Function to compute a single FWI gradient
    """
    u = TimeFunction(name='u', grid=model.grid,
                     space_order=order)
    grad = Function(name='grad', grid=model.grid)

    for i in nshots:
        # Update source location for each shot
        src.coordinates.data[0. :] =
            source_loc[i]

        # Run forward modelling operator
        op_fwd(u=u, src=src, rec=smooth_d)

        # Compute gradient from data residual and
        # update objective function
        residual = smooth_d.data[:] -
            true_d.data[:]
        objective +=
            .5*np.linalg.norm(residual)**2
        op_grad(rec=residual, u=u, m=model.m,
            grad=grad)

    return objective, grad.data
```

**Figure 7.** Definition of FWI gradient update.

```
model = Model(...)
dt, nt = <timestepping parameters>

# Define source and receiver geometry
src = RickerSource(...)
rec = Receiver(...)

# Create forward and gradient operators
op_fwd = forward(model, src, rec, order)
op_grad = gradient(model, rec, order)

# Run FWI with gradient descent
for i in range(0, fwi_iterations):
    # Compute functional value and gradient
    # for the current model estimate
    phi, direction = fwi_gradient(model.m)

    # Artificial Step length for gradient descent
    alpha = .005 / np.max(direction)

    # Update the model estimate and inforce
    # minimum/maximum values
    m_updated = model.m.data - alpha*direction
    model.m.data[:] = box_constraint(m_updated)
```

**Figure 8.** FWI algorithm with linesearch.

Devito in Fig. 8. With the forward and adjoint/gradient operator defined for a given source, we only need to add a loop over all the source experiments and the reduction operation on the gradients (sum the gradient for each source experiment together). In practice, this loop over sources is where the main task-based or MPI based parallelization happens. The wave-equation propagator does use some parallelization with multithreading or domain decomposition but that parallelism requires

5  communication. The parallelism over source experiment is task-based and does not require any communication between the separate tasks as the gradient for each source can be computed independently and reduced to obtain the full gradient. With the complete gradient summed over the source experiments, we update the model with a simple fixed step length gradient update (Cauchy, 1847).

This FWI function in Fig. 7 can then be included in any black-box optimization toolbox such as *SciPy* `optimize` to solve

10  the inversion problem Eq.6. While black-box optimization methods aim to minimize the objective, there are no guarantees they find a global minimum because the objective is highly non-linear in $m$ and other more sophisticated methods are required (Warner and Guasch, 2014; van Leeuwen and Herrmann, 2015; Peters and Herrmann, 2017; Witte et al., 2018a).

## 5    Verification

Given the operators defined in Sec. 3 we now verify the correctness of the code generated by the Devito compiler. We first verify that the discretized wave equation satisfies the convergence properties defined by the order of discretization, and secondly we verify the correctness of the discrete adjoint and computed gradient.

### 5.1    Numerical accuracy

The numerical accuracy of the forward modeling operator (Fig. 4) and the runtime achieved for a given spatial discretization order and grid size are compared to the analytical solution of the wave equation in a constant media. We define two measures of the accuracy that compare the numerical wavefield in a constant velocity media to the analytical solution:

- **Accuracy versus size**, where we compare the obtained numerical accuracy as a function of the spatial sampling size (grid spacing).

- **Accuracy versus time**, where we compare the obtained numerical accuracy as a function of runtime for a given physical model (fixed shape in physical units, variable grid spacing).

The measure of accuracy of a numerical solution relies on a hypothesis that we satisfy for these two tests:

- The domain is large enough and the propagation time small enough to ignore boundary related effects, i.e. the wavefield never reaches the limits of the domain.

- The source is located on the grid and is a discrete approximation of the Dirac to avoid spatial interpolation errors. This hypothesis guarantees the existence of the analytical and numerical solution for any spatial discretization (Igel, 2016).

### 5.1.1    Convergence in time

We analyze the numerical solution against the analytical solution and verify that the error between these two decreases at a second order rate as a function of the time step size $\Delta t$. The velocity model is a $400\text{m} \times 400\text{m}$ domain with a source at the center. We compare the numerical solution to the analytical solution on Fig. 9.

The analytical solution is defined as (Watanabe, 2015):

$$u_s(r,t) = \frac{1}{2\pi} \int\limits_{-\infty}^{\infty} \{-i\pi H_0^{(2)}(kr)q(\omega)e^{i\omega t}d\omega\} \tag{9}$$

$$r = \sqrt{(x-x_{src})^2 + (y-y_{src})^2}, \tag{10}$$

where $H_0^{(2)}$ is the Hankel function of second kind and $q(\omega)$ is the spectrum of the source function. As we can see on Fig. 10 the error decreases near quadratically with the size of the time step with a time convergence rate of slope of 1.94 in logarithmic scale that matches the theoretical expectation from a second order temporal discretization.

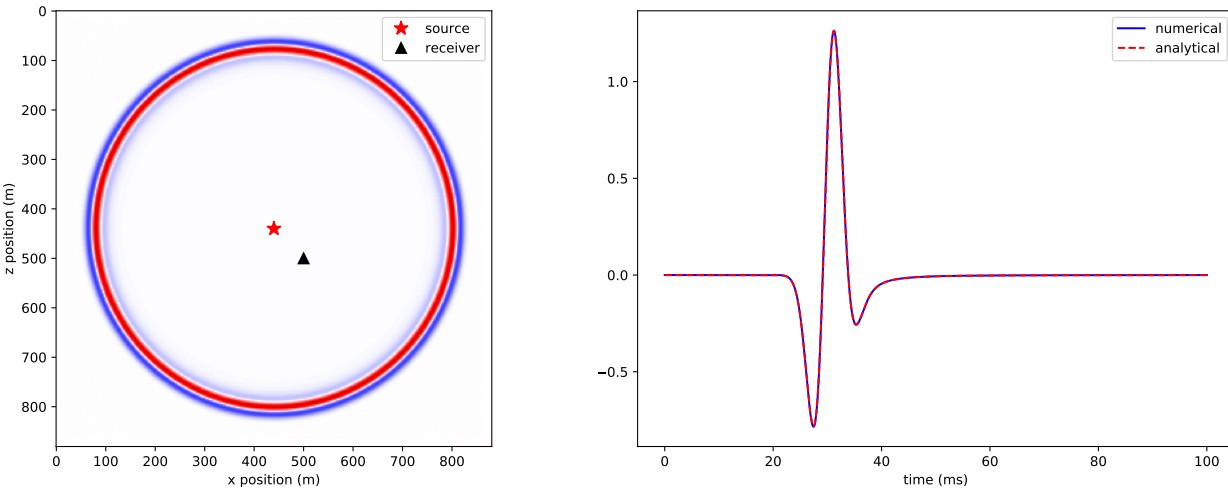

**Figure 9.** Numerical wavefield for a constant velocity $dt = .1$ms, $h = 1$m and comparison with the analytical solution.

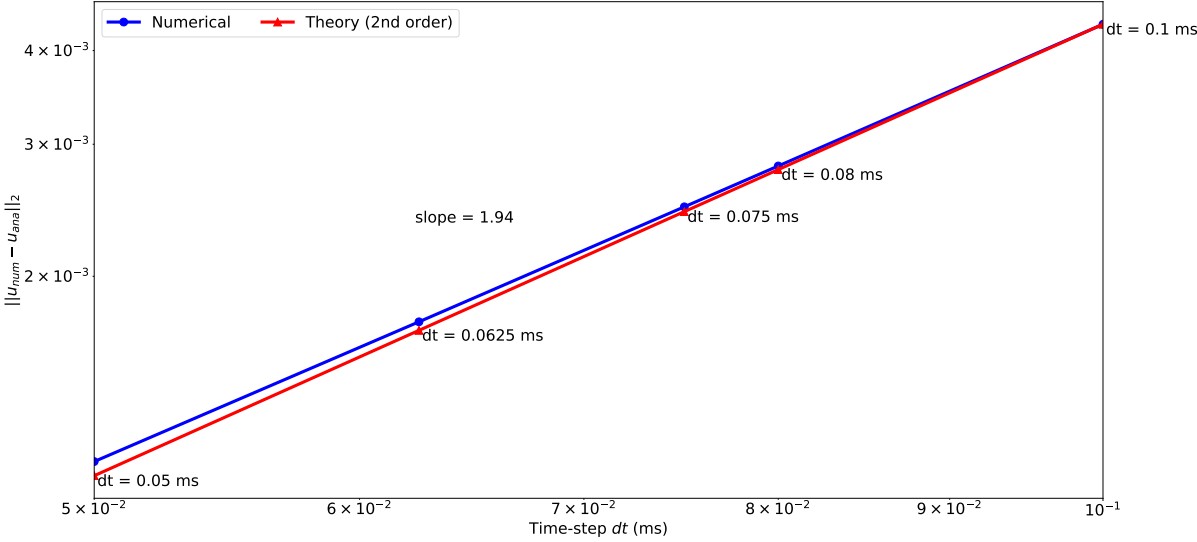

**Figure 10.** Time discretization convergence analysis for a fixed grid, fixed propagation time (150ms) and varying time step values. The result is plotted in a logarithmic scale and the numerical convergence rate (1.94 slope) shows that the numerical solution is accurate.

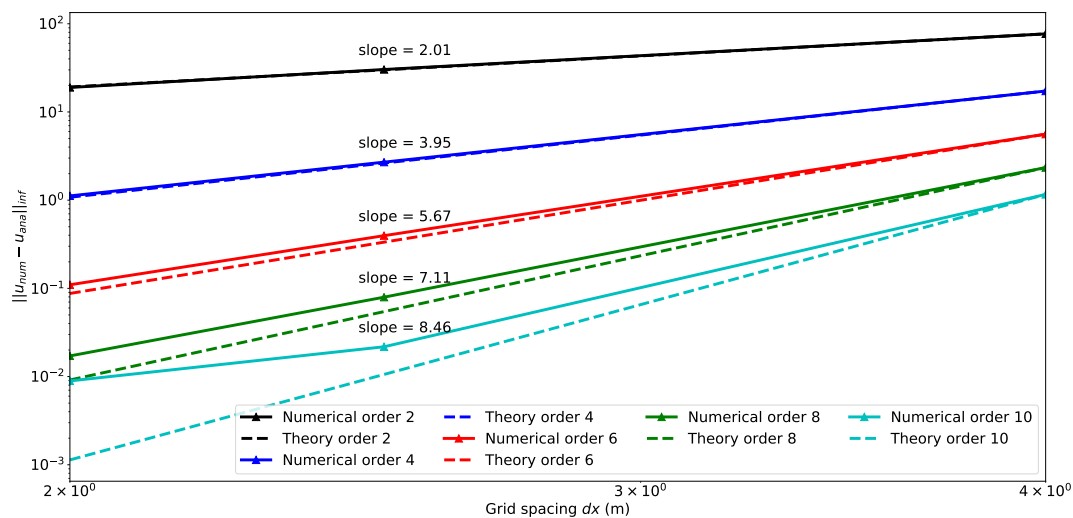

**Figure 11.** Comparison of the numerical convergence rate of the spatial finite difference scheme with the theoretical convergence rate from the Taylor theory. The theoretical rates are the dotted line with the corresponding colors. The result is plotted in a logarithmic scale to highlight the convergence orders as linear slopes and the numerical convergence rates show that numerical solution is accurate.

### 5.1.2 Spatial discretization analysis

The spatial discretization analysis follows the same method as the temporal discretixzation analysis. We model a wavefield for a fixed temporal setup with a small enough time-step to ensure negligeable time discretization error ($dt = .00625ms$). We vary the grid spacing ($dx$) and spatial discretization order and the and compute the error between the numerical and analytical solution. The convergence rates should follow the theoretical rates defined in Eq. 2. In details, for a $k^{th}$ order discretization in space, the error between the numerical and analytical solution should decrease as $O(dx^k)$. The best way to look at the convergence results is to plot the error in logarithmic scale and verify that the error decrease linearly with slope $k$. We show the convergence results on Fig. 11. The numerical convergence rates follow the theoretical ones for every tested order $k = 2, 4, 6, 8$ with the exception of the $10^{th}$ order for small grid size. This is mainly due to reaching the limits of the numerical accuracy and a value of the error on par with the temporal discretization error. This behavior for high order and small grids is however in accordance with the literature as in in Wang et al. (2017).

The numerical slopes obtained and displayed on Fig. 11 demonstrate that the spatial finite difference follows the theoretical errors and converges to the analytical solution at the expected rate. These two convergence results (time and space) verify the accuracy and correctness of the symbolic discretization with Devito. With this validated simulated wavefield, we can now verify the implementation of the operators for inversion.

| Order | $< \mathbf{F}\boldsymbol{x}, \boldsymbol{y} >$ | $< \boldsymbol{x}, \mathbf{F}^T\boldsymbol{y} >$ | relative error |
|---|---|---|---|
| 2nd order | 7.9858e+05 | 7.9858e+05 | 0.0000e+00 |
| 4th order | 7.3044e+05 | 7.3044e+05 | 0.0000e+00 |
| 6th order | 7.2190e+05 | 7.2190e+05 | 4.8379e-16 |
| 8th order | 7.1960e+05 | 7.1960e+05 | 4.8534e-16 |
| 10th order | 7.1860e+05 | 7.1860e+05 | 3.2401e-16 |
| 12th order | 7.1804e+05 | 7.1804e+05 | 6.4852e-16 |

**Table 1.** Adjoint test for different discretization orders in 2D, computed on a two layer model in double precision.

| Order | $< \mathbf{F}\boldsymbol{x}, \boldsymbol{y} >$ | $< \boldsymbol{x}, \mathbf{F}^T\boldsymbol{y} >$ | relative error |
|---|---|---|---|
| 2nd order | 5.3840e+04 | 5.3840e+04 | 1.3514e-16 |
| 4th order | 4.4725e+04 | 4.4725e+04 | 3.2536e-16 |
| 6th order | 4.3097e+04 | 4.3097e+04 | 3.3766e-16 |
| 8th order | 4.2529e+04 | 4.2529e+04 | 3.4216e-16 |
| 10th order | 4.2254e+04 | 4.2254e+04 | 0.0000e+00 |
| 12th order | 4.2094e+04 | 4.2094e+04 | 1.7285e-16 |

**Table 2.** Adjoint test for different discretization orders in 3D, computed on a two layer model in double precision.

## 5.2 Propagators verification for inversion

We concentrate now on two tests, namely the adjoint test (or dot test) and the gradient test. The adjoint state gradient of the objective function defined in Eq. 7 relies on the solutions of the forward and adjoint wave equations, therefore, the first mandatory property to verify is the exact derivation of the discrete adjoint wave equation. The mathematical test we use is the
standard adjoint property or dot-test:

for any random $\boldsymbol{x} \in \text{span}(\boldsymbol{P}_s\boldsymbol{A}(\boldsymbol{m})^{-T}\boldsymbol{P}_r^{-T}), \boldsymbol{y} \in \text{span}(\boldsymbol{P}_r\boldsymbol{A}(\boldsymbol{m})^{-1}\boldsymbol{P}_s^{-T})$

$$\frac{< \boldsymbol{P}_r\boldsymbol{A}(\boldsymbol{m})^{-1}\boldsymbol{P}_s^{-T}\boldsymbol{x}, \boldsymbol{y} > - < \boldsymbol{x}, \boldsymbol{P}_s\boldsymbol{A}(\boldsymbol{m})^{-T}\boldsymbol{P}_r^{-T}\boldsymbol{y} >}{< \boldsymbol{P}_r\boldsymbol{A}(\boldsymbol{m})^{-1}\boldsymbol{P}_s^{-T}\boldsymbol{x}, \boldsymbol{y} >} = 0.0. \tag{11}$$

The adjoint test is also individually performed on the source/receiver injection/interpolation operators in the Devito tests suite. The results, summarized in Tables 1 and 2 with $\mathbf{F} = \boldsymbol{P}_r\boldsymbol{A}(\boldsymbol{m})^{-1}\boldsymbol{P}_s^{-T}$, verify the correct implementation of the adjoint
operator for any order in both 2D and 3D. We observe that the discrete adjoint is accurate up to numerical precision for any order in 2D and 3D with an error of order $1e-16$. In combination with the previous numerical analysis of the forward modeling propagator that guarantees that we solve the wave equation, this result verifies that the adjoint propagator is the exact numerical adjoint of the forward propagator and that it implements the adjoint wave equation.

With the forward and adjoint propagators tested, we finally verify that the Devito operator that implements the gradient of
the FWI objective function (Eq. 7, Fig.6) is accurate with respect to the Taylor expansion of the FWI objective function. For a given velocity model and associated squared slowness $\boldsymbol{m}$, the Taylor expansion of the FWI objective function from Eq. 6 for a model perturbation $\boldsymbol{dm}$ and a perturbation scale $h$ is:

$$\Phi_s(\boldsymbol{m} + h\boldsymbol{dm}) = \Phi_s(\boldsymbol{m}) + \mathcal{O}(h)$$
$$\Phi_s(\boldsymbol{m} + h\boldsymbol{dm}) = \Phi_s(\boldsymbol{m}) + h\langle\nabla\Phi_s(\boldsymbol{m}), \boldsymbol{dm}\rangle + \mathcal{O}(h^2). \tag{12}$$

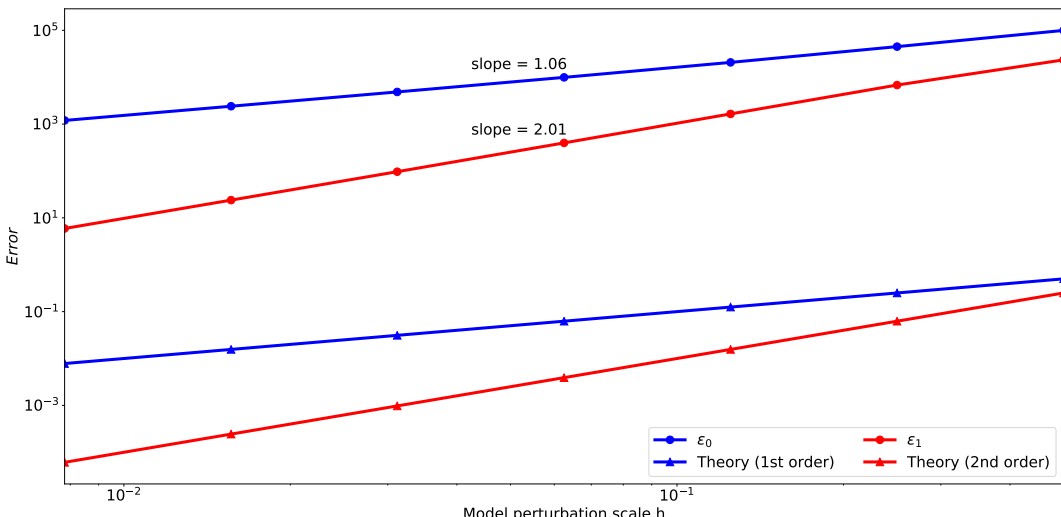

**Figure 12.** Gradient test for the acoustic propagator. The first order (blue) and second order (red) errors are displayed in logarithmic scales to highlight the slopes. The numerical convergence order (1.06 and 2.01) show that we have a correct implementation of the FWI operators.

These two equations constitute the gradient test where we define a small model perturbation $\boldsymbol{dm}$ and vary the value of $h$ between $10^{-6}$ and $10^0$ and compute the error terms:

$$\epsilon_0 = \Phi_s(\boldsymbol{m} + h\boldsymbol{dm}) - \Phi_s(\boldsymbol{m})$$
$$\epsilon_1 = \Phi_s(\boldsymbol{m} + h\boldsymbol{dm}) - \Phi_s(\boldsymbol{m}) - h\langle\nabla\Phi_s(\boldsymbol{m}), \boldsymbol{dm}\rangle. \tag{13}$$

We plot the evolution of the error terms as a function of the perturbation scale $h$ knowing $\epsilon_0$ should be first order (linear with slope 1 in a logarithmic scale) and $\epsilon_1$ should be second order (linear with slope 2 in a logarithmic scale). We executed the gradient test defined in Eq. 12 in double precision with a $8^{th}$ order spatial discretization. The test can be run for higher orders in the same manner but since it has already been demonstrated that the adjoint is accurate for all orders, the same results would be obtained.

In Fig. 12, the matching slope of the error term with the theoretical $h$ and $h^2$ slopes from the Taylor expansion verifies the accuracy of the inversion operators. With all the individual parts necessary for seismic inversion, we now validate our implementation on a simple but realistic example.

### 5.3    Validation: Full-Waveform Inversion

We show a simple example of FWI Eq. 7 on the Marmousi-ii model (Versteeg, 1994). This result obtained with the Julia
interface to Devito JUDI (Witte et al., 2018a; Witte et al., 2018) that provides high-level abstraction for optimization and linear algebra. The model size is $4\text{km} \times 16\text{km}$ discretized with a $10\text{m}$ grid in both directions. We use a $10\text{Hz}$ Ricker wavelet with $4\text{s}$ recording. The receivers are placed at the ocean bottom ($210\text{m}$ depth) every $10\text{m}$. We invert for the velocity with all the

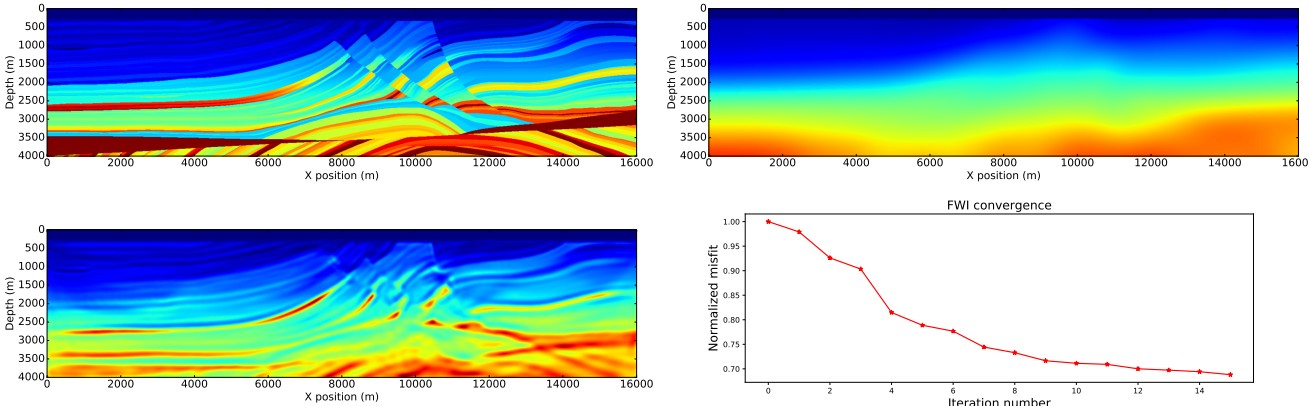

**Figure 13.** FWI on the acoustic Marmousi-ii model. The top-left plot is the true velocity model, the top-right is the initial velocity model, the bottom-left plot is the inverted velocity at the last iteration of the iterative inversion and the bottom-right plot is the convergence.

sources, spaced by $50\mathrm{m}$ at $10\mathrm{m}$ depth for a total of 300 sources. The inversion algorithm used is minConf_PQN(Schmidt et al., 2009), an l-BFGS algorithm with bounds constraints (minimum and maximum velocity values constraints). While conventional optimization would run the algorithm to convergence, this strategy is computationally not feasible for FWI. As each iteration requires two PDE solves per source $q_s$ (see adjoint state in Sec. 4), we can only afford a $\mathcal{O}(10)$ iterations in practice ($\mathcal{O}(10^4)$

PDE solves in total). In this example, we fix the number of function evaluations to 20, which, with the line search, corresponds to 15 iteration. The result is shown in Fig. 13 and we can see that we obtain a good reconstruction of the true model. More advanced algorithms and constraints will be necessary for more complex problem such as less accurate initial model, noisy data or field recorded data (Witte et al., 2018a; Peters and Herrmann, 2017); however the wave propagator would not be impacted, making this example a good proof of concept for Devito.

This result highlights two main contributions of Devito. First, we provide PDE simulation tools that allow easy and efficient implementation of inversion operator for seismic problem and potentially any PDE constrained optimization problem. As described in Sec. 3 and 4, we can implement all the required propagators and the FWI gradient in a few lines in a concise and mathematical manner. Second, as we obtained this results with JUDI (Witte et al., 2018b), a seismic inversion framework that provides a high-level linear abstraction layer on top of Devito for seismic inversion, this example illustrates that Devito is fully

compatible with external languages and optimizations toolboxes and allows users to use our symbolic DSL for finite difference within their own inversion framework.

## 5.4  Computational Fluid Dynamics

Finally we describe three classical computational fluid dynamics examples to highlight the flexibility of Devito for another application domain. Additional CFD examples can be found in the Devito code repository in the form of a set of Jupyter

notebooks. The three examples we describe here are the convection equation, the Burger equation and the Poisson equation.

```
u = TimeFunction(name='u', grid=grid)

# Derive stencil from symbolic equation
eq = Eq(u.dt + c*u.dxl + c*u.dyl)
stencil = solve(eq, u.forward)

# Apply boundary conditions
u.data[:, 0, :] = 1.
u.data[:, -1, :] = 1.
u.data[:, :, 0] = 1.
u.data[:, :, -1] = 1.

# Create an Operator that updates the forward stencil
# point in the interior subdomain only.
op = Operator(Eq(u.forward, stencil,
    subdomain=grid.interior))
```

**Figure 14.** Convection equation in Devito. In this example, the initial Dirichlet boundary conditions are set to 1 using the API indexing feature, which allows to assign values to the `TensorFunction` data.

These examples are adapted from Barba and Forsyth (2018) and the example repository contains both the original Python implementation with Numpy and the implementation with Devito for comparison.

### 5.4.1 Convection

The convection governing equation for a field $u$ and a speed $c$ in two dimensions is:

$$5 \quad \frac{\partial u}{\partial t} + c\frac{\partial u}{\partial x} + c\frac{\partial u}{\partial y} = 0. \tag{14}$$

The same way we previously described it for the wave equation, $u$ is then defined as a `TimeFunction`. In this simple case, the speed is a constant and does not need a symbolic representation, but a more general definition of this equation is possible with the creation of $c$ as a Devito `Constant` that can accept any runtime value. We then discretized the PDE using forward differences in time and backward differences in space:

$$10 \quad u_{i,j}^{n+1} = u_{i,j}^n - c\frac{\Delta t}{\Delta x}(u_{i,j}^n - u_{i-1,j}^n) - c\frac{\Delta t}{\Delta y}(u_{i,j}^n - u_{i,j-1}^n), \tag{15}$$

which is implemented in Devito as in Fig. 14.

The solution of the convection equation is displayed on Fig. 15 that shows the evolution of the field $u$ and the solution is consistent with the expected result produced by (Barba and Forsyth, 2018).

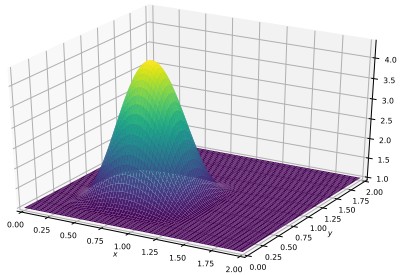 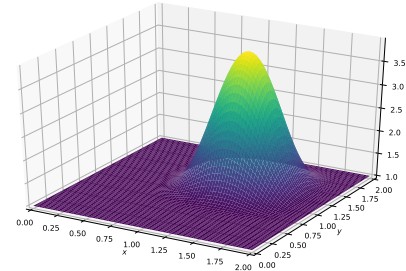

**Figure 15.** Initial (left) and final (right) time of the simulation of the convection equation.

### 5.4.2 Burgers' equation

In this second example, we show the solution of Burgers' equation. This example demonstrates that Devito supports coupled system of equations and non linear equations easily. The Burgers' equation in two dimensions is defined as the following coupled PDE system:

$$
\begin{cases}
\frac{\partial u}{\partial t} + u\frac{\partial u}{\partial x} + v\frac{\partial u}{\partial y} = \nu\left(\frac{\partial^2 u}{\partial x^2} + \frac{\partial^2 u}{\partial y^2}\right), \\
\frac{\partial v}{\partial t} + u\frac{\partial v}{\partial x} + v\frac{\partial v}{\partial y} = \nu\left(\frac{\partial^2 v}{\partial x^2} + \frac{\partial^2 v}{\partial y^2}\right),
\end{cases}
\tag{16}
$$

where $u, v$ are the two components of the solution and $\nu$ is the diffusion coefficient of the medium. The system of coupled equations is implemented in Devito in a few lines as shown in Fig. 16.

We show the initial state and the solution at the last time step of the Burgers' equation in Fig. 17. Once again, the solution corresponds to the reference solution of Barba and Forsyth (2018).

### 5.4.3 Poisson

We finally show the implementation of a solver for the Poisson equation in Devito. While the Poisson equation is not time dependent, the solution is obtained with an iterative solver and simplest one can easily be implemented with finite differences. The Poisson equation for a field $p$ and a right hand side $b$ is defined as:

$$
\frac{\partial^2 p}{\partial x^2} + \frac{\partial^2 p}{\partial y^2} = b,
\tag{17}
$$

and its solution can be computed iteratively with:

```python
# Define our velocity fields and initialise with hat
    function
u = TimeFunction(name='u', grid=grid, space_order=2)
v = TimeFunction(name='v', grid=grid, space_order=2)
init_hat(field=u.data[0], dx=dx, dy=dy, value=2.)
init_hat(field=v.data[0], dx=dx, dy=dy, value=2.)

# Write down the equations with explicit backward
    differences
a = Constant(name='a')
u_dx = first_derivative(u, dim=x, side=left, order=1)
u_dy = first_derivative(u, dim=y, side=left, order=1)
v_dx = first_derivative(v, dim=x, side=left, order=1)
v_dy = first_derivative(v, dim=y, side=left, order=1)
eq_u = Eq(u.dt + u*u_dx + v*u_dy, a*u.laplace,
    subdomain=grid.interior)
eq_v = Eq(v.dt + u*v_dx + v*v_dy, a*v.laplace,
    subdomain=grid.interior)

# Let SymPy rearrange our stencils to form the update
    expressions
stencil_u = solve(eq_u, u.forward)
stencil_v = solve(eq_v, v.forward)
update_u = Eq(u.forward, stencil_u)
update_v = Eq(v.forward, stencil_v)

# Create Dirichlet BC expressions using the low-level
    API
bc_u = [Eq(u[t+1, 0, y], 1.)] # left
bc_u += [Eq(u[t+1, nx-1, y], 1.)] # right
bc_u += [Eq(u[t+1, x, ny-1], 1.)] # top
bc_u += [Eq(u[t+1, x, 0], 1.)] # bottom
bc_v = [Eq(v[t+1, 0, y], 1.)] # left
bc_v += [Eq(v[t+1, nx-1, y], 1.)] # right
bc_v += [Eq(v[t+1, x, ny-1], 1.)] # top
bc_v += [Eq(v[t+1, x, 0], 1.)] # bottom

# Create the operator
op = Operator([update_u, update_v] + bc_u + bc_v)
```

**Figure 16.** Burgers' equations in Devito. In this example, we use explicitly the FD function `first_derivative`. This function provides more flexibility and allows to take an upwind derivative, rather than a standard centered derivative ($\dot{d}x$), to avoid odd-even coupling, which leads to chessboard artifacts in the solution.

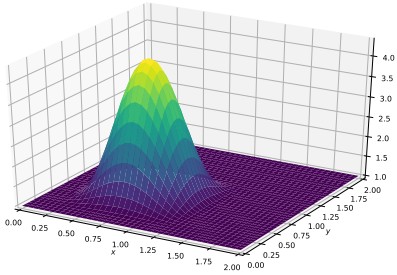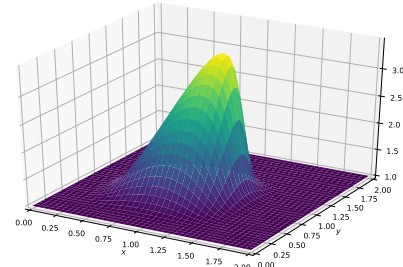

**Figure 17.** Initial (left) and final (right) time of the simulation of the Burgers' equations.

$$p_{i,j}^{n+1} = \frac{(p_{i+1,j}^n + p_{i-1,j}^n)\Delta y^2 + (p_{i,j+1}^n + p_{i,j-1}^n)\Delta x^2 - b_{i,j}^n \Delta x^2 \Delta y^2}{2(\Delta x^2 + \Delta y^2)}, \tag{18}$$

where the expression in Eq. 18 is computed until either the number of iterations is reached (our example case) or more realistically when $||p_{i,j}^{n+1} - p_{i,j}^n|| < \epsilon$. We show two different implementations of a Poisson solver in Fig. 18, 19. While these two implementations produce the same result, the second one takes advantage of Devito's `BufferedDimension` that allows to iterate automatically alternating between $p^n$ and $p^{n+1}$ as the two different time buffers in the `TimeFunction`.

The solution of the Poisson equation is displayed on Fig. 20 with its right-hand-side $b$.

These examples demonstrate the flexibility of Devito and show that a broad range of PDE can easily be implemented with Devito including non linear equation, coupled PDE system and steady state problems.

## 6  Performance

In this section we demonstrate the performance of Devito from the numerical and the inversion point of view, as well as the absolute performance from the hardware point of view. This section only provides a brief overview of Devito's performance and a more detailed description of the compiler and its performance is covered in (Luporini et al., 2018).

### 6.1  Error-cost analysis

Devito's automatic code generation lets users define the spatial and temporal order of FD stencils symbolically and without having to reimplement long stencils by hand. This allows users to experiment with trade-offs between discretization errors and runtime, as higher order FD stencils provide more accurate solutions that come at increased runtime. For our error-cost analysis, we compare absolute error in $L_2$-norm between the numerical and the reference solution to the time-to-solution (the numerical and reference solution are defined in the previous Sec. 5). Fig. 21 shows the runtime and numerical error obtained

```
p = Function(name='p', grid=grid, space_order=2)
pd = Function(name='pd', grid=grid, space_order=2)
p.data[:] = 0.
pd.data[:] = 0.

# Initialise the source term 'b'
b = Function(name='b', grid=grid)
b.data[:] = 0.
b.data[int(nx / 4), int(ny / 4)] = 100
b.data[int(3 * nx / 4), int(3 * ny / 4)] = -100

# Create Laplace equation base on 'pd'
eq = Eq(pd.laplace, b, subdomain=grid.interior)
# Let SymPy solve for the central stencil point
stencil = solve(eq, pd)
# Now we let our stencil populate our second buffer 'p'
eq_stencil = Eq(p, stencil)

# Create boundary condition expressions
x, y = grid.dimensions
t = grid.stepping_dim
bc = [Eq(p[x, 0], 0.)]
bc += [Eq(p[x, ny-1], 0.)]
bc += [Eq(p[0, y], 0.)]
bc += [Eq(p[nx-1, y], 0.)]

# Now we can build the operator that we need
op = Operator([eq_stencil] + bc)

# Run the outer loop explicitly in Python
for i in range(nt):
    # Determine buffer order
    if i % 2 == 0:
        _p = p
        _pd = pd
    else:
        _p = pd
        _pd = p

    # Apply operator
    op(p=_p, pd=_pd)
```

**Figure 18.** Poisson equation in Devito with field swap in Python.

```
# Now with Devito we will turn 'p' into 'TimeFunction'
# object to make all the buffer switching implicit
p = TimeFunction(name='p', grid=grid, space_order=2)

# Initialise the source term 'b'
b = Function(name='b', grid=grid)
b.data[:] = 0.
b.data[int(nx / 4), int(ny / 4)] = 100
b.data[int(3 * nx / 4), int(3 * ny / 4)] = -100

# Create Laplace equation base on 'p'
eq = Eq(p.laplace, b)
# Let SymPy solve for the central stencil point
stencil = solve(eq, p)
# Let our stencil populate the buffer 'p.forward'
eq_stencil = Eq(p.forward, stencil)

# Create boundary condition expressions
# Note that we now add an explicit "t + 1"
# for the time dimension.
bc = [Eq(p[t + 1, x, 0], 0.)]
bc += [Eq(p[t + 1, x, ny-1], 0.)]
bc += [Eq(p[t + 1, 0, y], 0.)]
bc += [Eq(p[t + 1, nx-1, y], 0.)]

# We can even switch performance logging back on,
# since we only require a single kernel invocation.
configuration['log-level'] = 'INFO'

# Create and execute the operator for nt iterations
op = Operator([eq_stencil] + bc)
op(time=nt)
```

**Figure 19.** Poisson equation in Devito with buffered dimension for automatic swap at each iteration.

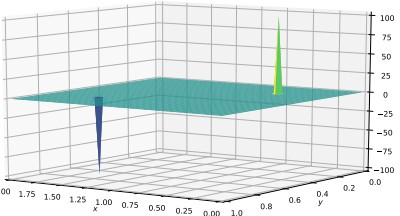 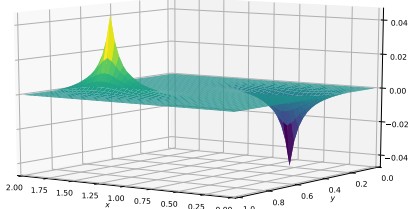

**Figure 20.** Right hand side (left) and solution (right) of the Poisson equations

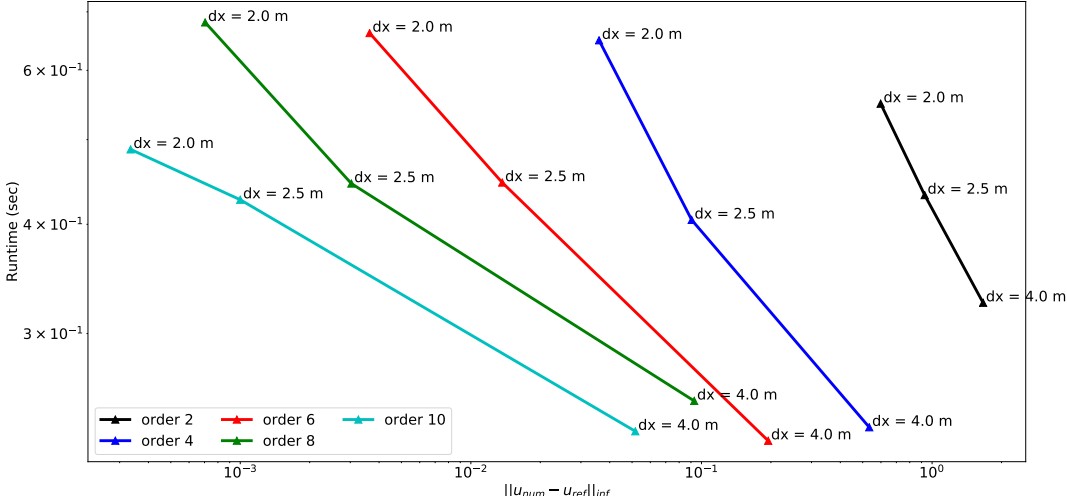

**Figure 21.** Different spatial discretization orders accuracy against runtime for a fixed physical setup (model size in m and propagation time).

for a fixed physical setup. We use the same parameter as in Sec. 5.1 with a domain of $400\text{m} \times 400\text{m}$ and we simulate the wave propagation for $150\text{ms}$.

The results in Fig. 21 illustrate that higher order discretizations produce a more accurate solution on a coarser grid with a smaller runtime. This result is very useful for inverse problems, as a coarser grid requires less memory and fewer time steps. A grid size two times bigger implies a reduction of memory usage by a factor of $2^4$ for 3D modeling. Devito then allows users to design FD simulators for inversion in an optimal way, where the discretization order and grid size can be chosen according to the desired numerical accuracy and availability of computational resources. While a near linear evolution of the runtime with increasing space order might be expected, we do not see such a behavior in practice. The main reason for this, is that the effect of Devito's performance optimizations for different space orders is difficult to predict and does not necessarily follow a linear

relationship. On top of these optimizations, the runtimes also include the source injection and receiver interpolation, which impact the runtime in a non-linear way. Therefore these results are still acceptable.. The order of the FD stencils also affects the best possible hardware usage that can theoretically be achieved and whether an algorithm is compute or memory bound, a trade-off that is described by the roofline model.

## 6.2   Roofline analysis

We present performance results of our solver using the roofline model, as previously discussed in Colella (2004); Asanovic et al. (2006); Patterson and Hennessy (2007); Williams et al. (2009); Louboutin et al. (2017a). Given a finite difference scheme, this method provides an estimate of the best achievable performance on the underlying architecture, as well as an absolute measure of the hardware usage. We also show a more classical metric, namely *time to solution*, in addition to the roofline plots, as both are essential for a clear picture of the achieved performance. The experiments were run on an Intel Skylake 8180 architecture (28 physical cores, 38.5 MB shared L3 cache, with cores operating at 2.5 Ghz). The observed Stream TRIAD (McCalpin, 1991-2007) was 105 GB/s. The maximum single-precision FLOP performance was calculated as $\#cores \cdot \#avx\ units \cdot \#data\ items\ per\ vector\ register \cdot 2(fused\ multiply-add) \cdot core\ frequency = 4480\ \mathrm{GFLOPs/s}$. A (more realistic) performance peak of 3285 GFLOPs/s was determined by running the LINPACK benchmark (Dongarra, 1988). These values are used to construct the roofline plots. In the performance results presented in this section, the operational intensity (OI) is computed by the Devito profiler from the symbolic expression after the compiler optimization. While the theoretical OI could be used, we chose to recompute it from the final optimized symbolic stencil for a more accurate performance measure. A more detailed overview of Devito's performance model is described in Luporini et al. (2018).

We show three different roofline plots, one plot for each domain size attempted, in Fig. 22, 23 and 24. Different space orders are represented as different data points. The time-to-solution in seconds is annotated next to each data point. The experiments were run with all performance optimizations enabled. Because auto-tuning is used at runtime to determine the optimal loop-blocking structure, timing only commences after autotuning has finished. The reported operational intensity benefits from the use of expression transformations as described in Sec. 3; particularly relevant for this problem is the factorization of FD weights.

We observe that the time to solution increases nearly linearly with the size of the domain. For example, for a 16th order discretization, we have a 17.1sec runtime for a $512 \times 512 \times 512$ domain and 162.6sec runtime for a $1024 \times 1024 \times 1024$ domain (8 times bigger domain and about 9 times slower). This is not surprising: the computation lies in the memory-bound regime and the working sets never fit in the L3 cache. We also note a drop in performance with a 16th order discretization (relative to both the other space orders and the attainable peak), especially when using larger domains (Fig. 23 and 24). Our hypothesis, supported by profiling with Intel VTune (Intel Corporation, 2016), is that this is due to inefficient memory usage, in particular misaligned data accesses. Our plan to improve the performance in this regime consists of resorting to a specialized stencil optimizer such as YASK (see Sec. 7). These results show that we have a portable framework that achieves good performance on different architectures. There is small room for improvements, as the machine peak is still relatively distant, but 50-60%

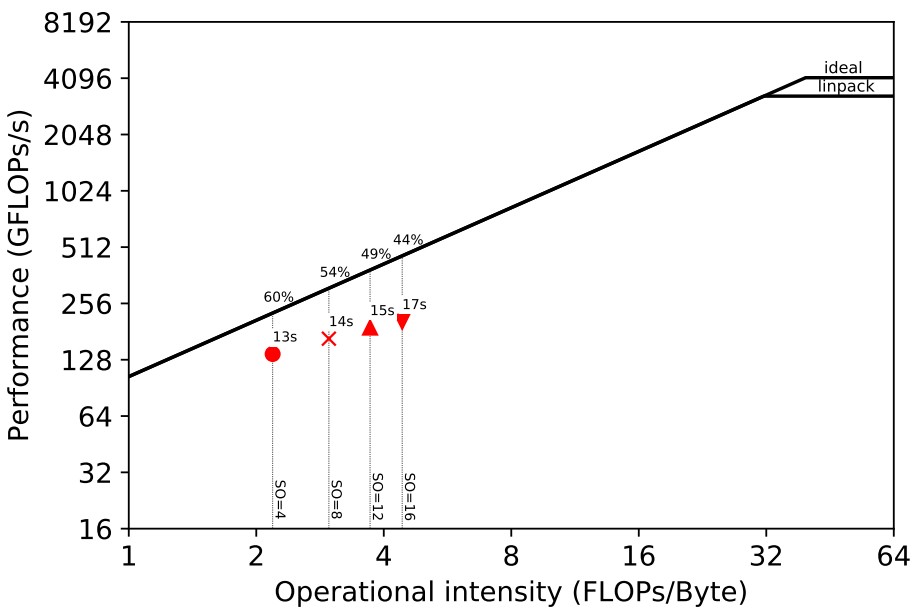

**Figure 22.** Roofline plots for a $512 \times 512 \times 512$ model on a Skylake 8180 architecture. The run times correspond to 1000ms of modeling for four different spatial discretization orders (4, 8, 12, 16).

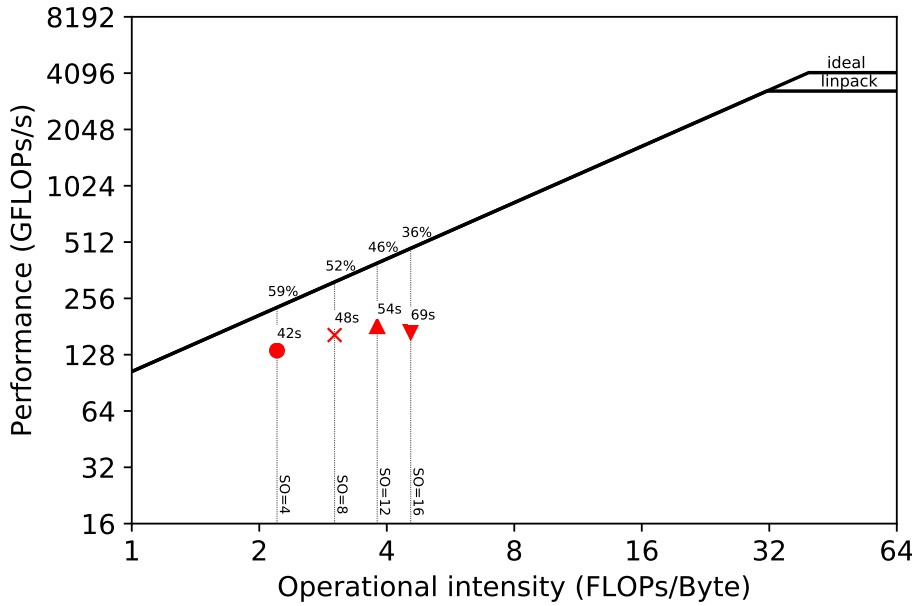

**Figure 23.** Roofline plots for a $768 \times 768 \times 768$ model on a Skylake 8180 architecture. The run times correspond to 1000ms of modeling for four different spatial discretization orders (4, 8, 12, 16).

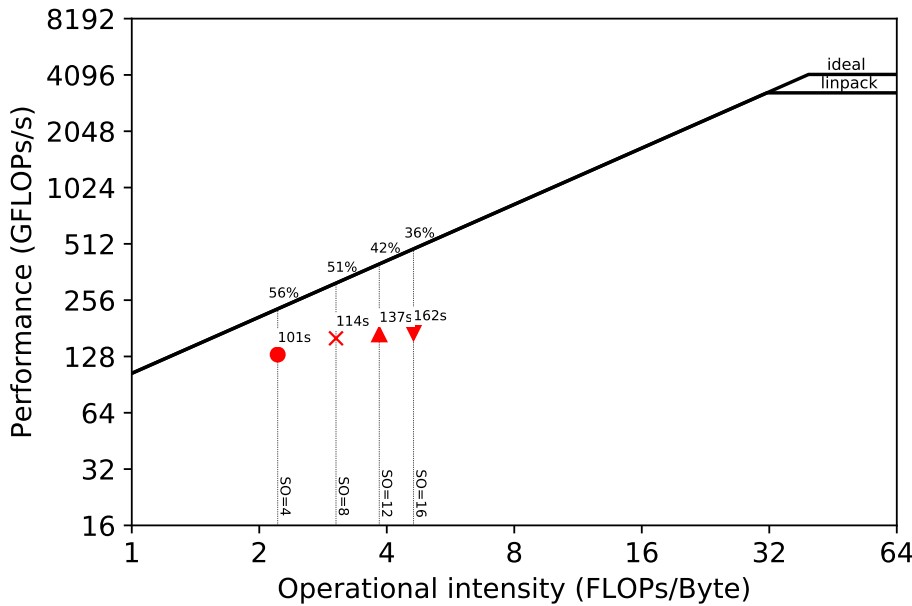

**Figure 24.** Roofline plots for a $1024 \times 1024 \times 1024$ model on a Skylake 8180 architecture. The run times correspond to 1000ms of modeling for four different spatial discretization orders (4, 8, 12, 16).

of the attainable peak is usually considered very good. Finally, we remark that testing on new architectures will only require extensions to the Devito compiler, if any, while the application code remains unchanged.

## 7 Future Work

A key motivation for developing an embedded DSL such as Devito is to enable quicker development, simpler maintenance, and better portability and performance of solvers. The other benefit of this approach is that HPC developer effort can be focused on developing the compiler technology that is reapplied to a wide range of problems. This software reuse is fundamental to keep the pace of technological evolution. For example, one of the current projects in Devito regards the integration of YASK (Yount, 2015), a lower-level stencil optimizer conceived by Intel for Intel architectures. Adding specialized backends such as YASK – meaning that Devito can generate and compile YASK code, rather than pure C/C++ – is the key for long-term performance portability, one of the goals that we are pursuing. Another motivation is to enable large-scale computations and as many different types of PDEs as possible. In practice, this means that a staggered grid setup with half-node discretization and domain decomposition will be required. These two main requirements to extend the DSL to a broader community and applications are in full development and will be made available in future releases.

## 8 Conclusions

We have introduced a DSL for time-domain simulation for inversion and its application to a seismic inverse problem based on the finite difference method. Using the Devito DSL a highly optimized and parallel finite difference solver can be implemented within just a few lines of Python code. Although the current application focuses on features required for seismic imaging applications, Devito can already be used in problems based on other equations; a series of CFD examples are included in the code repository.

The code traditionally used to solve such problems is highly complex. The primary reason for this is that the complexity introduced by the mathematics is interleaved with the complexity introduced by performance engineering of the code to make it useful for practical use. By introducing a separation of concerns, these aspects are decoupled and both simplified. Devito successfully achieves this decoupling while delivering good computational performance and maintaining generality, both of which shall continue to be improved in future versions.

## 9 Code Availability

The code source code, examples and test script are available on github at https://github.com/opesci/devito and contains a README for installation. A more detailed overview of the project, list of publication and documentation of the software generated with Sphinx is available at http://www.devitoproject.org/. To install Devito:

```
git clone -b v3.1.0 https://github.com/opesci/devito
cd devito
conda env create -f environment.yml
source activate devito
```

```
pip install -e .
```

*Acknowledgements.* The development of Devito was primarily supported through the Imperial College London Intel Parallel Computing Centre. We would also like to acknowledgement the support from the SINBAD II project and support of the member organizations of the SINBAD Consortium as well as EPSRC grants EP/M011054/1 and EP/L000407/1.

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
