# Peer review of "Devito (v3.1.0): an embedded domain-specific language for finite differences and geophysical exploration"

_Geoscientific Model Development, 2018_

## Referee Comment (RC1) · Anonymous Referee #1 · 27 Sep 2018

In their paper "Devito (v3.1.0): an embedded domain-specific language for finite differences and geophysical exploration" the authors present a software that introduces a high level representation for partial differential equations discretized with finite difference. The paper presents how this software can be used to solve wave equations and apply it to Full wave inversion problems. These problems are typical for geophysical exploration and thematically fits the GMD. The authors does a thorough verification of the full wave inversion implementation, which raises the quality of the paper. However, I would advise the authors to revise section 3.2 and 4, as it lacks references to previously published results. Also, I advice the authors to show the applicability of the software by presenting multiple examples with different partial differential equations.

Major issues:

Section 3.2 and Section 4 explain how the domain specific language Devito is structured. However, these sections do not refer to [1-3], which are tutorials published in The Leading Edge that presents the Devito implementation of the problem. I suggest the authors include these references, as well as revise these chapters, as some of the details in this section can be excluded.

The paper present Marmousi-ii model. This is a very interesting example, which is very relevant for showing the applicability of the software. However, their description of the optimization setup is insufficient. I suggest the authors add information about the number of iterations needed, the stopping criterion and the initial and final functional values.

As mentioned in the conclusion, Devito can be applied to other equations, as CFD problems. I would like the authors to present one or two CFD examples to emphasis the ease of use and generality of Devito.

Minor Issues:

Page 7, Figure 2: Missing period after figure caption

Page 7, Line 9: This formula does not makes sense for me. For k=2, this is a sum from 1 to 1, which excludes the midpoint and conflicts with the first equation on Page 8. Also k=3 would make it a sum from 1 to 1.5.

Page 8, Figure 3: The text in between the code and equivalent output makes this look like a part of the text. Please add borders around figure to clarify that this is one figure. Especially unfortunate since this is mid sentence from the previous page.

Page 11, Line 1: \delta d missing subscript s.

Page 11, Line 7: Three sentences in a row starts with "We have".

Page 12, Figure 4: The variable "src" is not defined explicitly in this paper. However, it

is defined in [1].

Page 16, Line 1: "..convergence results is yo" should be "is to"

Page 17, Line 12: Replace comma after equation (11) with a period

Page 21, Line 16: Reference not properly formatted

Page 22, Figures 16-18: have too small fontsize. I also think all these figures can be combined

References:

[1] Louboutin, Mathias, et al. "Full-waveform inversion, Part 1: Forward modeling." The Leading Edge 36.12 (2017): 1033-1036. https://library.seg.org/doi/pdfplus/10.1190/tle36121033.1

[2] Louboutin, Mathias, et al. "Full-waveform inversion, Part 2: Adjoint modeling." The Leading Edge 37.1 (2018): 69-72. https://library.seg.org/doi/pdf/10.1190/tle37010069.1

[3] Witte, Philipp, et al. "Full-waveform inversion, Part 3: Optimization." The Leading Edge 37.2 (2018): 142-145. https://library.seg.org/doi/pdfplus/10.1190/tle37020142.1

---

## Referee Comment (RC2) · Anonymous Referee #2 · 27 Nov 2018

In this paper **"Devito (v3.1.0): an embedded domain-specific language for finite differences and geophysical exploration"** has been introduced. The authors present the new version of their numerical tool which discretises the governing equations written at a higher level and describes how it can be used to solve the full waveform inversion problems, which fits in the scope of the journal. Although the paper is very well written some major descriptions are missing from the manuscript, which reduces the overall quality of the paper. I advise the authors to revise some of the claims they made.

Major Issues:

1. Page 5: lines 10-15, the authors claim that "Devito is compatible with a wide range of tools available in the Python software stack", but it is not clear if this can be done on-the-fly using the operator function as devito produces a low-level C99 code. In short how the data is managed between C and Python is not clear, if file I/O is used then how is it advantageous?
2. Page 6: lines 1-2, the authors claim that CSE is used as an optimisation technique, as this is based on SymPy's CSE the reference given from 2015 is obsolete as the CSE capabilities of SymPy has changed a lot after that. Also, specifically the authors should address the following
   a. CSE of SymPy considers the function arguments as sub-expressions, as all the operators are based on "Function" class how this is handled?
   b. Also, the authors should specifically mention which version of SymPy is Devito compatible with
3. Page 6: line 10, "Devito provides two symbolic object types that mimic SymPy symbols, enabling the construction of stencil expressions in symbolic form", this statement needs justification as the next lines the authors claim to use "sympy function" which is not of type symbol, these two points contradict each other from SymPy point of view.
4. Page 6: line 28, "TimeFunction" is it derived from "Funciton" class given in line 13 of page 6?, this should be clear in the manuscript
5. As mentioned in the conclusions, such a framework can be applied to CFD problems. More examples should be given and I feel that only OpenMP parallelisation reduces the problem sets that can be solved for CFD, this should be clearly mentioned. Also, authors should provide a comment on the half-node interpolation capabilities of the framework as this is essential for most CFD cases

Minor issues:

1. Type-setting fractions appearing in text should be inlined this should be implemented during type-setting stage
2. Page 16: line 3 it should be $10^{th}$ and not 10th
3. Page 21: line 16 reference is not proper
4. The text in figures 16-18 are too small to read on print.
5. Page 8: Figure 3 is confusing due to lack of borders
6. The full form of the acronym FWI is repeated at two places this should be corrected.
7. Figures 16-18 How the operational intensity is evaluated?

---

## Author Comment (AC1) · 19 Dec 2018

We thank you for your review time and very useful review to make this paper impactful. We answered all the reviewer requests and discuss our choices on revision changes. Please find attached below response to the review and attached the revised manuscript with the updates in blue.

Major Issues:

1. Page 5: lines 10-15, the authors claim that "Devito is compatible with a wide range of tools available in the Python software stack", but it is not clear if this can

be done on-the-fly using the operator function as devito produces a low-level C99 code. In short how the data is managed between C and Python is not clear, if file I/O is used then how is it advantageous?

- The memory is managed by numpy. What we mean is that even though we generate low level c99 code, the arrays are all numpy array that are accessible and modifiable in Python at any time. If file I/O is used, it would be in python and read into a numpy array or written from the numpy array, not within the C code.

2. Page 6: lines 1-2, the authors claim that CSE is used as an optimisation technique, as this is based on SymPy's CSE the reference given from 2015 is obsolete as the CSE capabilities of SymPy has changed a lot after that. Also, specifically the authors should address the following

    a. CSE of SymPy considers the function arguments as sub-expressions, as all the operators are based on "Function" class how this is handled?
    b. Also, the authors should specifically mention which version of SymPy is Devito compatible with

The Sympy reference is updated to most recent one, we changed it to the "cite us" for the sympy github repo. - a. Added sentence to precise that we do not use Sympy CSE but have a custom implementation of it. - b. Added version.

3. Page 6: line 10, "Devito provides two symbolic object types that mimic SymPy symbols, enabling the construction of stencil expressions in symbolic form", this statement needs justification as the next lines the authors claim to use "sympy function" which is not of type symbol, these two points contradict each other from SymPy point of view.

- A symbolic expression is not necessarly a Symbol. While Sympy does the distinction between a Symbol and a Function, any Sympy expression is a symbolic expression (sympy = symbolic python).

4. Page 6: line 28, "TimeFunction" is it derived from "Function" class given in line 13 of page 6?, this should be clear in the manuscript

- Added explanation of TimeFunction inheritance and extra parameters such as time_order for the time discretization and save for the size of the time axis if the full history is saved.

5. As mentioned in the conclusions, such a framework can be applied to CFD problems. More examples should be given and I feel that only OpenMP parallelisation reduces the problem sets that can be solved for CFD, this should be clearly mentioned. Also, authors should provide a comment on the half-node interpolation capabilities of the framework as this is essential for most CFD cases

- Added 3 CFD examples that highlight the flexibility of Devito. More example are available in the repository and this is now emphasized in the manuscript. The domain decomposition (MPI) and half-node FD are discussed.

Minor issues: 1. Type-setting fractions appearing in text should be inlined this should be implemented during type-setting stage

- Fixed

2. Page 16: line 3 it should be 10th and not 10th

    • Fixed

3. Page 21: line 16 reference is not proper

    • Fixed

4. The text in figures 16-18 are too small to read on print.

    • Font size increased in figures for readability

5. Page 8: Figure 3 is confusing due to lack of borders

    • Border added to all listings for consistency

6. The full form of the acronym FWI is repeated at two places this should be corrected.

    • Fixed

7. Figures 16-18 How the operational intensity is evaluated?

- Added explanation

Please also note the supplement to this comment:
https://www.geosci-model-dev-discuss.net/gmd-2018-189/gmd-2018-189-AC1-supplement.pdf

**Supplement:**

[revised manuscript text omitted]

---

## Author Comment (AC2) · 19 Dec 2018

We thank you for your review time and very useful review to make this paper impactful. We answered all the reviewer requests and discuss our choices on revision changes. Please find attached below response to the review and attached the revised manuscript with the updates in blue.

In their paper "Devito (v3.1.0): an embedded domain-specific language for finite differences and geophysical exploration" the authors present a software that introduces a high level representation for partial differential equations discretized with finite difference. The paper presents how this software can be used to solve wave equations and

apply it to Full wave inversion problems. These problems are typical for geophysical exploration and thematically fits the GMD. The authors does a thorough verification of the full wave inversion implementation, which raises the quality of the paper. However, I would advise the authors to revise section 3.2 and 4, as it lacks references to previously published results. Also, I advice the authors to show the applicability of the software by presenting multiple examples with different partial differential equations.

Major issues:

Section 3.2 and Section 4 explain how the domain specific language Devito is structured. However, these sections do not refer to [1-3], which are tutorials published in The Leading Edge that presents the Devito implementation of the problem. I suggest the authors include these references, as well as revise these chapters, as some of the details in this section can be excluded.

- We do understand why these section may seem a bit lengthy for someone who already read the TLE tutorials, we think that TLE is a very domain specific journal and that a lot of GMD readers will not have read these. We therefore think it is better to keep this section detailed and self contained rather than shorten it and refer to a journal people may not know.

The paper present Marmousi-ii model. This is a very interesting example, which is very relevant for showing the applicability of the software. However, their description of the optimization setup is insufficient. I suggest the authors add information about the number of iterations needed, the stopping criterion and the initial and final functional values.

- A convergence plot and details about the experimental set-up have been added to the manuscript.

**[GMDD](about:blank)**
As mentioned in the conclusion, Devito can be applied to other equations, as CFD problems. I would like the authors to present one or two CFD examples to emphasis the ease of use and generality of Devito.

- Added 3 CFD examples that highlight the flexibility of Devito. More examples are available in the repository and this is now mentioned in the manuscript. The domain decomposition (MPI) and half-node FD are discussed.

Minor Issues:

Page 7, Figure 2: Missing period after figure caption

- Fixed

Page 7, Line 9: This formula does not makes sense for me. For k=2, this is a sum from 1 to 1, which excludes the midpoint and conflicts with the first equation on Page 8. Also k=3 would make it a sum from 1 to 1.5.

- The sum runs from k=0 to floor(k/2) (0 to 1 for k=2,3) and fixed accordingly. Odd orders do not exist except first order so order k=5 would be the same as order k=4. changed accordingly

Page 8, Figure 3: The text in between the code and equivalent output makes this look like a part of the text. Please add borders around figure to clarify that this is one figure. Especially unfortunate since this is mid sentence from the previous page.

- Added borders

Page 11, Line 1: $\delta$ d missing subscript s.

• Fixed

Page 11, Line 7: Three sentences in a row starts with "We have".

• Fixed

Page 12, Figure 4: The variable "src" is not defined explicitly in this paper. However, it is defined in [1].

• Added reference and brief note on it.

Page 16, Line 1: "..convergence results is yo" should be "is to"

• Fixed

Page 17, Line 12: Replace comma after equation (11) with a period

• Fixed

Page 21, Line 16: Reference not properly formatted

• Fixed

Page 22, Figures 16-18: have too small fontsize. I also think all these figures can be combined

• Combined

References: [1] Louboutin, Mathias, et al. "Full-waveform inversion, Part 1: Forward modeling." The Leading Edge 36.12 (2017): 1033-1036. https://library.seg.org/doi/pdfplus/10.1190/tle36121033.1 [2] Louboutin, Mathias, et al. "Full-waveform inversion, Part 2: Adjoint modeling." The Leading Edge 37.1 (2018): 69-72. https://library.seg.org/doi/pdf/10.1190/tle37010069.1 [3] Witte, Philipp, et al. "Full-waveform inversion, Part 3: Optimization." The Leading Edge 37.2 (2018): 142-145. https://library.seg.org/doi/pdfplus/10.1190/tle37020142.1

Please also note the supplement to this comment:
https://www.geosci-model-dev-discuss.net/gmd-2018-189/gmd-2018-189-AC2-supplement.pdf

---

## Author Response (AR1)

**Author response**

Mathias Louboutin

*Correspondence to:* Mathias Louboutin (mlouboutin3@gatech.edu)

**1 Editor request**

your manuscript looks good and the review process will start once you add a very slight change to your title. At GMD it is common practice that the title contains the model name and the version number. So you may add "1.0"

**2 Author response**

5 The manuscript title has been updated with the version of the release associated with the manuscript (v3.1.0) and the installation instruction as well to install that specific version. Regards

---

## Referee Report (RR1)

In their revised version of the paper **"Devito (v3.1.0): an embedded domain-specific language for finite differences and geophysical exploration"** the authors have additional CFD-examples to emphasize the generality of their software. Additionally, they have fixed the minor issues that was raised in the first round of revisions.

I accept the authors argumentation for a self contained section about the implementation, and I am pleased to see that they now refer to their previously published work on the subject in *The Leading Edge*.

***Technical corrections:***
*Page 3, Line 3:* Missing space in the added "and computational fluid dynamics".
*Page 20, Line 1:* Missing space after *minConf_PQN*
*Page 20, Line 19:* "Convection" should be non-capitalized.
*Figures 14-17, 20:* Missing punctuation at the end of caption.

***Minor issues:***
*Page 20, Line 3:* Doesn't each iteration require a single PDE solve per source $q_s$? Please rewrite this sentence to make it clearer.
*Page 20, Line 4:* We can only afford a (10) … is not a complete sentence. Please revise.
*Page 21, Figure 14:* Caption is a bit too short. I suggest adding a comment about the Dirichlet BCs, as they are not mentioned in the text.
*Page 23, Figure 16:* Caption is too short. I would add a comment regarding *first_derivative* and how it differs from *dx*, which has been used in the previous sections.
*Section 5.4.3:* Instead of using point sources for the Poisson problem and visualizing them, I suggest using the *Method of Manufactured Solutions* to shown that the Devito implementation is correct.

---

## Author Response (AR2)

We thank the reviewer for their insight and useful comments on our manuscript. We made the required changes to the manuscript and you can find below our answer to the comments

**Reviewer**

In their revised version of the paper "Devito (v3.1.0): an embedded domain-specific language for finite differences and geophysical exploration" the authors have additional CFD-examples to emphasize the generality of their software. Additionally, they have fixed the minor issues that was raised in the first round of revisions.

I accept the authors argumentation for a self contained section about the implementation, and I am pleased to see that they now refer to their previously published work on the subject in The Leading Edge.

Technical corrections:

1. Page 3, Line 3: Missing space in the added "and computational fluid dynamics".

- fixed

2. Page 20, Line 1: Missing space after minConf_PQN

- fixed

3. Page 20, Line 19: "Convection" should be non-capitalized.

- fixed

4. Figures 14-17, 20: Missing punctuation at the end of caption.

- fixed

Minor issues:

1. Page 20, Line 3: Doesn't each iteration require a single PDE solve per source $q_s$? Please rewrite this sentence to make it clearer.

- Each iteration does require two PDE solves not one as explained in the FWI section. A first one to model the synthetic data $d_{syn}$ and the forward wavefield $u_s$, then a second one to solve the adjoint wave equation to compute the gradient. This is conventional adjoint state formulation that requires two PDE to compute the gradient. The sentence now refers to the theory section for clarity.

2. Page 20, Line 4: We can only afford a (10) ... is not a complete sentence. Please revise.

- Sentence is revised for clarity.

3. Page 21, Figure 14: Caption is a bit too short. I suggest adding a comment about the Dirichlet BCs, as they are not mentioned in the text.

- Caption extended with Dirichlet BC explanation.

4. Page 23, Figure 16: Caption is too short. I would add a comment regarding first_derivative and how it differs from dx, which has been used in the previous sections.

- Caption extended with explanation of the difference with the conventional `.dx`

5. Section 5.4.3: Instead of using point sources for the Poisson problem and visualizing them, I suggest using the Method of Manufactured Solutions to shown that the Devito implementation is correct.

- We do agree that MMS would improve the verification of the implementation. However, the integration and implementation of MMS in and for Devito is currently in progress and not available in the version presented in this Paper. However, as we explained in the paper, all part of Devito are tested including the finite differences against polynomial (for which FD is exact for a properly chosen order) and are available in the `test` subfolder of Devito.

**Editor**

Dear authors

Moreover, I have some comments.

Regarding the content:

1. I have a question on Fig. 21. If I had connected all symbols with the same dx (not the same order as done here) in order to show the dependence of runtime on order, I would have expect to see monotonically decreasing curves. Is the numerical time step fixed? For dx=2m for instance, runtime increases slightly up to order 8, but the the order 10 simulation is faster. So the general finding on p24l16 is not really correct. Could you elaborate on this and modify the manuscript accordingly.

- In this figure, the time step is defined by the CFL condition and therefore it is bigger for dx=4m than for dx=2m (roughly dt is doubled), but the time step is fixed for a constant grid spacing. However, the runtimes take a lot of different parameters into account, which are not all listed in the manuscript. A complete overview of the Devito compiler paper has been submitted independently and is referenced in the performance section. Overall, the compiler performs optimizations such as factorizations, flop reductions, etc., that will lead to non-monotonic runtime variations. Secondly, the solver contains source and receiver interpolations that impact the performance in a non-linear way. Finally, as the roofline model shows, higher order stencils allow better usage of the hardware (better ratio of FLOPs/sec) and will perform better in some cases than lower orders, due to better flop vs memory traffic.
- Clarifications have been added to the paragraph.

2. If you want you can include the alternative verification approach proposed by the reviewer concerning the Poisson equation. I agree with the reviewer, that Fig.20 is no formal verification, however I think that main intent of the new application examples is to show the versatility of Devito. In any case, future Devito users have to take care, that for their specific problem, Devito produces reliable solutions.

- Indeed, as you pointed out, this example mainly shows the versatility of Devito adding CFD examples as requested by the previous reviewer. While a more proper verification, such as MMS, would improve this example, we do not have it implemented at the moment and for the version of Devito presented here. This is however a work in progress and will be properly explained in a follow up paper that will also include staggered grid and domain decomposition, two points that were raised by one of the reviewer.

Regarding language etc:

1. p13l8: "on its library Revolve", what does "its" refer to?

- Rephrased for clarity

2. You sometimes miss the parentheses around the citations, e.g. p2l22, p3l33 or opposite way p20l20

- Citation fixed.

3. I spotted several typos: p2l29, p3l3, p8l15, p12l10: Fig., p15l21, p17l11, p18l12: remove "we"?, p20l4 (10): missing O?, p20l8: "the PDE solve part"?, p22l5 "corresponds to" = "consistent with"?, p27l13 This list is quite long and probably not complete, hence I expect that you go over the whole manuscript to find further typos.

- Removed lots of typos.

4. What about the Witte-manuscript ".. Part 3 Optimization". Has it been accepted in the meantime? The provided URL does not look like a link that will live.

- This paper is accepted and the bibtex entry is updated to the journal url doi.

Best wishes, Simon Unterstraßer